# CAR: Conceptualization-Augmented Reasoner for Zero-Shot Commonsense Question Answering

**Weiqi Wang[1][*], Tianqing Fang[1][*], Wenxuan Ding[1], Baixuan Xu[1], Xin Liu[1],**
**Yangqiu Song[1], Antoine Bosselut[2]**

[1]Department of Computer Science and Engineering, HKUST, Hong Kong SAR, China
[2]NLP Lab, School of Computer and Communication Sciences, EPFL, Switzerland
{wwangbw, tfangaa, yqsong}@cse.ust.hk, antoine.bosselut@epfl.ch

## Abstract

The task of zero-shot commonsense question answering evaluates models on their capacity to reason about general scenarios beyond those presented in specific datasets. Existing approaches for tackling this task leverage external knowledge from CommonSense Knowledge Bases (CSKBs) by pre-training the model on synthetic QA pairs constructed from CSKBs. In these approaches, negative examples (distractors) are formulated by *randomly* sampling from CSKBs using fairly primitive keyword constraints. However, two bottlenecks limit these approaches: the inherent incompleteness of CSKBs limits the semantic coverage of synthetic QA pairs, and the lack of human annotations makes the sampled negative examples potentially uninformative and contradictory.

To tackle these limitations above, we propose **C**onceptualization-**A**ugmented **R**easoner (CAR), a zero-shot commonsense question-answering framework that fully leverages the power of *conceptualization*. Specifically, CAR abstracts a commonsense knowledge triple to many higher-level instances, which increases the coverage of the CSKB and expands the ground-truth answer space, reducing the likelihood of selecting false-negative distractors. Extensive experiments demonstrate that CAR more robustly generalizes to answering questions about zero-shot commonsense scenarios than existing methods, including large language models, such as GPT3.5 and ChatGPT. Our code, data, and model checkpoints are available at https://github.com/HKUST-KnowComp/CAR.

## 1 Introduction

Pre-trained Language Models (PLMs; Devlin et al., 2019; Clark et al., 2020) fine-tuned on task-specific training sets achieve remarkable near-human performance on held-out test sets, yet struggle to generalize to examples that are distributionally different

* Equal Contribution

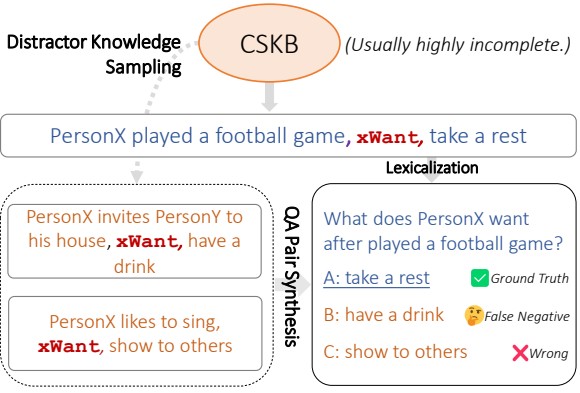

Figure 1: An example of constructing synthetic QA pairs from CSKB (Ma et al., 2021). The simple heuristic used in this process can result in false negative options.

from their training sets (McCoy et al., 2019; Ma et al., 2019; Zhou et al., 2021; Wang et al., 2021). This discrepancy arises because fine-tuned PLMs often rely on spurious, dataset-specific correlations to learn a task rather than learning to fully leverage implicit commonsense knowledge required for reasoning (Branco et al., 2021). For reasoning systems to be effective, though, they must be robust across domains and generalize beyond the specificities of individual datasets.

To confront the generalization issue in commonsense reasoning tasks, the task of zero-shot commonsense Question-Answering (QA) requires models to answer questions for evaluation benchmarks without access to their corresponding training data (Shwartz et al., 2020; Li et al., 2020). Among several methods that tackle this task, the most performant ones inject commonsense knowledge from CSKBs (Hwang et al., 2021; Jiang et al., 2021) into PLMs by fine-tuning them on synthetic QA pairs transformed from commonsense knowledge triples, where the head and relation are transformed to a question, and the tail serves as a ground answer. Negative examples are randomly sampled with keyword-overlap constraints (Ma et al., 2021). Such knowledge injection benefits not only QA

**Original Knowledge Triple**

(PersonX played a football game, **xWant,** take a rest)

**Conceptualization Relations**

(played a football game, **IsA,** *Sport*)
(played a football game, **IsA,** *Tiring event*)
...
(played a football game, **IsA,** *Exercise*)

**Conceptualization Inference**

$(h, r, t) \wedge (h, \texttt{IsA}, h_c)$

$\downarrow$

$(h_c, r, t)$

**Abstract Knowledge**

(PersonX (do) *Sport*, **xWant,** take a rest)
(PersonX (do) *Tiring event*, **xWant,** take a rest)
...
(PersonX (do) *Exercise*, **xWant,** take a rest)

Figure 2: An example of conceptualization inference. More abstracted knowledge, such as (Do sport, xWant, take a rest), can be obtained through conceptualization.

tasks that are derived from CSKBs, such as SocialIQA (Sap et al., 2019b), which is derived from ATOMIC (Sap et al., 2019a), but also QA datasets in other domains (Bisk et al., 2020).

Despite recent advancements in this area, two major challenges remain. First, manually curated CSKBs, such as ATOMIC, are incomplete (Kuo and Hsu, 2010). While consolidating multiple CSKBs can improve coverage, it remains infeasible to cover all conceivable knowledge for the vast range of entities and situations in the real world (He et al., 2022). Automatic methods for expanding CSKBs exist, such as knowledge base completion (Li et al., 2016; Malaviya et al., 2020), and knowledge distillation from large language models (West et al., 2022; Gao et al., 2023), but they either fail to provide knowledge about novel entities or only provide highly accurate yet less informative knowledge (e.g., vague adjectives, such as *happy*, as situation descriptors). Second, in zero-shot commonsense QA, negative examples are required for models to learn to distinguish the validity of commonsense scenarios (Chen et al., 2023a). However, existing negative QA examples are synthesized using simple heuristic-based negative sampling without considering deeper semantics, resulting in too many false negative options. For instance, in Figure 1, "have a drink" is also plausible in the context of "after playing a football game." These questions that label plausible options as negative instances confuse the model during training, impeding its ability to discern correct commonsense knowledge.

We tackle both of these challenges by utilizing *conceptualization*. As Murphy (2004) posits, humans rely on conceptual induction to draw inferences about unseen situations without the need for

memorizing specific knowledge. Conceptualization (He et al., 2022) offers a similar capability by abstracting a set of instances into concepts, which allows for the derivation of abstract commonsense knowledge associated with each concept that can be instantiated to assist reasoning on specific downstream situations. For example, in Figure 2, "play a football game" can be conceptualized as a *tiring event*, which further generalizes as abstract knowledge. The benefits of conceptualization are twofold. First, conceptualized commonsense knowledge introduces abstract knowledge through a one-step concept inference based on the original CSKB, enhancing knowledge coverage. Second, as the abstract knowledge is conditioned on the original knowledge, the *recall* of knowledge regarding the same head is increased, leading to more fine-grained constraints for negative option sampling.

Inspired by these advantages, we propose **CAR** (**C**onceptualization-**A**ugmented **R**easoner), a simple yet effective zero-shot commonsense QA framework that leverages *conceptualization* to expand existing CSKBs and reduce false-negative distractors. We first augment the original CSKB with conceptualization to infuse abstract commonsense knowledge to improve knowledge coverage. Then, we propose a conceptualization-constraint sampling strategy that generates distractors with concept-level constraints to prevent false negative options (Section 4). Experimental results on five popular commonsense QA benchmarks demonstrate the effectiveness of CAR, which even surpasses GPT3.5 and ChatGPT (Section 5). In Section 6, we analyze why CAR works by providing human evaluations that show a significant reduction of false negative options compared to other methods. Finally, our analysis reveals that conceptualization-augmented training examples tend to be more *ambiguous* (Swayamdipta et al., 2020) than those produced by prior heuristics, leading to better out-of-domain generalization.

## 2 Related Works

**Zero-shot Commonsense QA.** Zero-shot commonsense QA evaluates a model's reasoning generalizability on unseen QA entries without any supervision signals from the corresponding annotated training data. To tackle this task, two primary pipelines have emerged in existing works. The first paradigm employs off-the-shelf language models without changing the parameters, either using

vanilla language modeling with prompts (Trinh and Le, 2018; Li et al., 2022), or with some inference-time mechanisms specifically designed for reasoning, such as self-talk (Shwartz et al., 2020), cloze translation (Dou and Peng, 2022), and dynamic generation of reasoning sub-graphs and graph reasoning (Bosselut et al., 2021). The second pipeline leverages external CSKBs as knowledge sources to provide PLMs with additional supervision signals for further fine-tuning (Banerjee and Baral, 2020; Ma et al., 2021; Su et al., 2022). A common strategy involves converting knowledge triples in CSKBs to synthetic QA pairs by transforming the head and relation to a question, the tail to a gold answer, and (randomly) sample tails from other heads as distractors. Such fine-tuning paradigm benefits from incorporating CSKBs within different domains (Kim et al., 2022; Shi et al., 2023) and exploiting multi-hop graph structures with graph neural networks (Guan et al., 2023), and heightens the model's commonsense sensitivity in a QA context, which leads to state-of-the-art performances.

**Conceptualization.** Conceptualization refers to the process of abstracting a group of instances or events into a general concept (Song et al., 2011, 2015). In commonsense reasoning, it simulates conceptual induction (Murphy, 2004) and enables the derivation of abstract commonsense knowledge under the specific contextualization of the original commonsense knowledge (Tenenbaum et al., 2011), which is often lacking in existing CSKBs. Around many existing works studying conceptualization (Durme et al., 2009; Gong et al., 2016; Liu et al., 2022; Peng et al., 2022), He et al. (2022) investigate it at event-level semantics and construct AbstractATOMIC, an event conceptualization benchmark and knowledge base based on ATOMIC (Sap et al., 2019a). Recently, Wang et al. (2023a) propose to conceptualize CSKBs at scale with semi-supervised learning and demonstrate abstract knowledge can enhance commonsense inference modeling (Bosselut et al., 2019; Da et al., 2021). With current works mostly investigating the problem of *conceptualization* itself, none of them have extrinsically evaluated the impact of conceptualization on downstream tasks, such as commonsense QA (Talmor et al., 2019) or machine reading comprehension (Nguyen et al., 2016).

**Data Augmentation.** Data augmentation aims at generating new examples from existing data to expand the size and diversity of a training set without requiring costly data annotations (Wei and Zou, 2019). Various methods have been proposed to augment textual data, including those using random perturbation (Wei and Zou, 2019), text embeddings (Wang and Yang, 2015), lexical semantics (Niu and Bansal, 2018), back translation (Sennrich et al., 2016), and large language models (West et al., 2022; Ismayilzada and Bosselut, 2023; Gao et al., 2023) for CSKB construction. Nevertheless, text-perturbation-based augmentations do not provide new knowledge to CSKBs, and knowledge mining from large language models suffers from high typicality (e.g., favoring simple commonsense over informative yet rare commonsense) and low density, still making negative sampling subject to false negatives (Malaviya et al., 2020).

# 3 Problem Definition

## 3.1 Definitions

**Conceptualization.** Formally, denote a CSKB as $D$ with knowledge triples in the format of $D = \{(h, r, t) | h \in H, r \in R, t \in T\}$, where $H$, $R$, and $T$ are the sets of heads, relations, and tails in the original CSKB. Following He et al. (2022), the conceptualized CSKB, conditioned on $D$, can be denoted as $D^C = \{(h_c, r, t) | h_c \in H_c, r \in R, t \in T\}$, where $H_c$ is the set of conceptualized head events. Specifically, each conceptualized head $h_c$ is obtained by replacing a component $i \in h$ with its abstract concept $c$ while ensuring that the formed $(h_c, r, t)$ triple is still plausible in the original context $(r, t)$. Such $(h_c, r, t)$ triples are commonly referred to as abstract commonsense knowledge.

**Zero-shot Commonsense QA.** In this paper, we employ the zero-shot commonsense QA task proposed by Ma et al. (2021) to study our framework. First, the CSKB $D$ is transformed into multiple $(Q_i, A_i)$ pairs where $Q_i$ is a natural language question and $A_i = \{A_{i,1}, A_{i,2}, ..., A_{i,m}\}$ is a set of options with $m$ candidates. Specifically, for a given knowledge triple $(h, r, t) \in D$, we convert $h, r$ into $Q_i$ via natural language templates and use $t$ as the ground answer. Additionally, we retrieve $m - 1$ distractors from other triples sampled from $D$ using a manually defined strategy, such as keyword overlap filtering. The objective of our task is to train a QA model from the synthetic QA sets $D^Q = \{(Q_i, A_i) | (h_i, r_i, t_i) \in D\}$. Once trained, the model is tested on held-out test

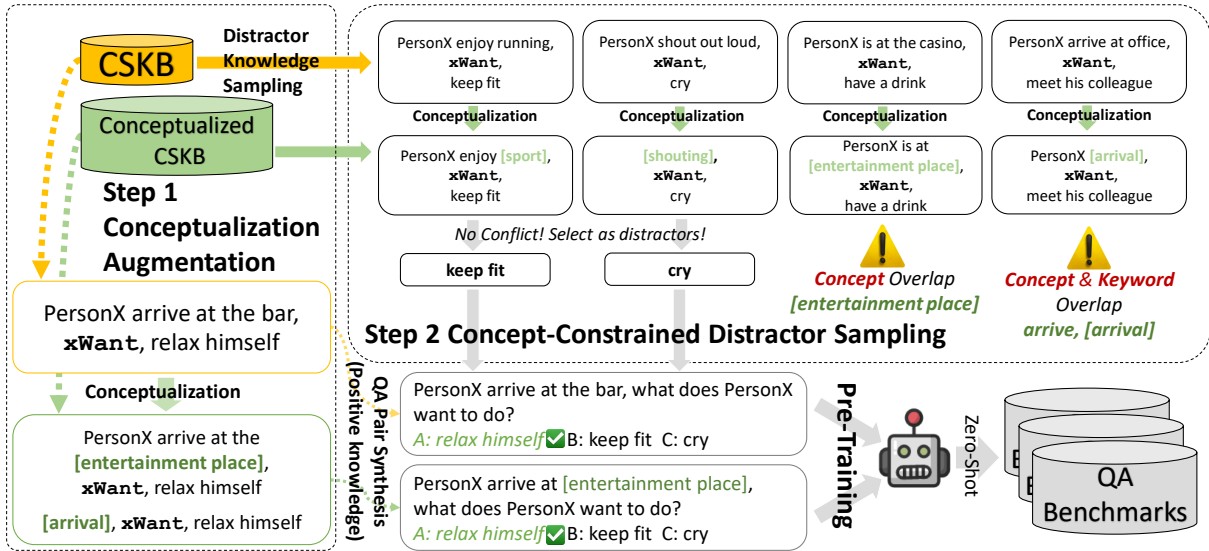

Figure 3: An overview of the CAR framework, which shows the process of synthesizing (PersonX arrive at the bar, xWant, relax himself) into QA pairs. The triple is conceptualized first, and potential distractor triples are sampled and filtered by keyword and concept overlap. Only those triples that have no overlap are used as distractors.

entries $(Q^{test}, A^{test})$ from QA benchmarks. This requires the model to perform zero-shot commonsense reasoning since the training data from the target benchmarks are unavailable to the model.

## 3.2 Dataset

We use ATOMIC (Sap et al., 2019b) as the source CSKB $D$. ATOMIC contains inferential commonsense knowledge, in the format of $(h, r, t)$ triple, that is associated with commonly seen events. Specifically, the heads of ATOMIC triples are events, whereas the tail nodes are either events or attributes. For conceptualization, we use the human-annotated abstract knowledge from AbstractATOMIC (He et al., 2022) to train a generative conceptualizer for acquiring $D^C$. More details of conceptualizations and statistics of AbstractATOMIC are provided in Section 4.1 and Appendix B.1.

## 3.3 Evaluation Benchmarks

Following Ma et al. (2021), we evaluate our framework on the validation split of five commonsense QA benchmarks: Abductive NLI (aNLI; Bhagavatula et al., 2020), CommonsenseQA (CSQA; Talmor et al., 2019), PhysicalIQA (PIQA; Bisk et al., 2020), SocialIQA (SIQA; Sap et al., 2019b), and WinoGrande (WG; Sakaguchi et al., 2021). These manually constructed benchmarks evaluate various knowledge types essential for robust commonsense reasoning (Kim et al., 2022). Detailed statistics and explanations of these benchmarks are provided in Appendix A.

## 4 CAR Framework

This section introduces our proposed CAR framework. A general sketch is presented in Figure 3. Our framework can be summarized into three steps: (1) Conduct one-step conceptualization inference on existing triples in the CSKB to obtain abstract commonsense knowledge triples. (2) Transfer the triples into QA pairs and generate distractors using keywords and conceptualizations as constraints. (3) Train the QA model using marginal ranking loss.

## 4.1 Conceptualization Augmentation

To incorporate abstract knowledge into the CSKB, we begin by augmenting the $(h, r, t) \in D$ triples by conducting a one-step conceptualization inference. Initially, given a head event $h$, we retrieve all plausible conceptualizations $C_h = \{c_{i_1,1}, c_{i_1,2}, ...\}$ for all identified instances $i \in \{i_1, i_2, ...|i \in h\}$ using entity-linking heuristics to retrieve concepts from Probase (Wu et al., 2012) and WordNet (Miller, 1995). The conceptualized head event $h_c$ is then obtained by replacing an $i \in h$ with one of its retrieved conceptualization $c \in \{c_{i,1}, c_{i,2}, ...\}$. This is done for all identified instances and their retrieved conceptualizations, thereby constructing the set of conceptualized head events of $h$. Subsequently, we link the non-abstract counterpart $(r, t)$ after $h_c$ to generate candidate abstract knowledge triples $(h_c, r, t)$, where we adopt a discriminator trained with a semi-supervised conceptualization-instantiation framework to determine their plausibility (Wang et al., 2023a). Only plausible triples

are kept to form $D^C$. Details about the conceptualization retrieval processes and the discriminator are presented in Appendix B.1.

## 4.2 Concept-Constrained QA Synthesis

To synthesize a commonsense triple $(h, r, t)$ into a $(Q_i, A_i)$ pair, we first transfer $h, r$ into $Q_i$ by using natural language templates and set $t$ as the ground-truth answer $A_1$. For example, the triple in Figure 3 becomes "PersonX arrives at the bar, what does PersonX want to do?" with the answer being "relax himself." Additional distractors are generated by transforming sampled distractor triples from the original CSKB, where only triples with the same commonsense relation $r$ are sampled to ensure informativeness. To prevent sampling false negative options, we constrain sampling distractor knowledge by filtering keywords and conceptualizations. Formally, denote the keywords of a head event $h$ as $T_h = \{t_1, t_2, \cdots\}$ and the full set of plausible conceptualizations for all identified instances in $h$ as $C_h = \{c_{i_1,1}, c_{i_1,2}, \cdots, c_{i_2,1}, \cdots\}$, we associate a triple $(h, r, t)$ with $T_h + C_h$ to form its constraint. Only knowledge triple $(h', r, t')$ which satisfies $(T_{h'} + C_{h'}) \cap (T_h + C_h) = \emptyset$ can be sampled as a distractor candidate. This constraint requires that the two triples have no common keywords, and their instances cannot be abstracted into the same conceptualization. For example, in Figure 3, "(PersonX is at the casino, xWant, have a drink)" cannot be used as a distractor triple because "casino" can be conceptualized as "entertainment place," which is the same as "bar" in the original triple. Finally, we sample two distractor triples for the triple $(h, r, t)$ and use the tails of these two triples as the distractors. To guarantee that the abstract commonsense knowledge from our previous augmentation is learnable by the QA model, we synthesize both the original triple $(h, r, t)$ and its conceptualized versions $(h_c, r, t)$ into QA pairs.

## 4.3 Model Training

Following Ma et al. (2021), we train our QA model by fine-tuning a pre-trained Masked Language Model (MLM) using the Marginal Ranking (MR) loss. Let $C$ represent the original context (if any), $Q$ represent the question, and $(A_1, A_2, ...)$ be the list of options. We first concatenate $C$, $Q$, and an answer option $A_i$ together via natural language prompts to generate input sequences $(T_1, T_2, ...)$. For example, the synthesized question with its correct answer in Figure 3 will be transformed as:

"PersonX arrives at the bar, as a result, PersonX want to, relax himself." We then repeatedly mask out a token at one time and calculate the masked loss. The final MLM score for an input sequence $T \in \{T_1, T_2, ...\}$ with $n$ tokens is:

$$\mathcal{S}(T) = -\frac{1}{n} \sum_{i=1}^{n} \log P(t_i | ..., t_{i-1}, t_{i+1}, ...) \quad (1)$$

After calculating the scores $S_1, S_2, ...$ for all answer candidates $A_1, A_2, ...$, we compute the marginal ranking loss based on Equation 2, where $\eta$ represents the margin and $y$ is the index of the correct answer.

$$\mathcal{L} = \frac{1}{m} \sum_{i=1, i \neq y}^{m} \max(0, \eta - S_y + S_i) \quad (2)$$

During the evaluation phase, we use the same scoring procedure to assign a score to each option and select the one whose concatenated sentence achieves the lowest score as the model's prediction.

## 5 Experiments

### 5.1 Setup

**Baselines** First, we use random voting (*Random*) and most-frequent labeling (*Majority*) to demonstrate the characteristics of each benchmark. Vanilla RoBERTa-Large (Liu et al., 2019), and DeBERTa-v3-Large (He et al., 2023) PLMs are used to demonstrate the power of fine-tuning. The performances of these two models under a supervised training regime are also included to show the upper bound of our results. We also include the results of several existing approaches that tackle the same task, including Self-talk (Shwartz et al., 2020), COMET-DynaGen (Bosselut et al., 2021), SMLM (Banerjee and Baral, 2020), MICO (Su et al., 2022), and the previous state-of-the-art STL-Adapter (Kim et al., 2022). Most importantly, we compare our framework with Ma et al. (2021) to validate the efficacy of conceptualization since both methods share similar model architecture and training procedures. Both RoBERTa-Large and DeBERTa-v3-Large are used as the backbones for fair comparisons. There are, in total, 534,833 synthetic QA pairs provided by Ma et al. (2021).

With the recent advances in Large Langauge Models (LLMs) (Bang et al., 2023; Chan et al., 2023; Qin et al., 2023), we also benchmark the performances of GPT3.5 (Brown et al., 2020) and ChatGPT (OpenAI, 2022) as baselines. We prompt

| Model | CSKB | a-NLI | CSQA | PIQA | SIQA | WG | Avg. |
|---|---|---|---|---|---|---|---|
| Random | - | 50.0 | 20.0 | 50.0 | 33.3 | 50.0 | 40.7 |
| Majority | - | 50.8 | 20.9 | 50.5 | 33.6 | 50.4 | 41.2 |
| RoBERTa-L (Liu et al., 2019) | - | 65.5 | 45.0 | 67.6 | 47.3 | 57.5 | 56.6 |
| DeBERTa-v3-L (He et al., 2023) | - | 59.9 | 25.4 | 44.8 | 47.8 | 50.3 | 45.6 |
| Self-talk (Shwartz et al., 2020) | - | - | 32.4 | 70.2 | 46.2 | 54.7 | - |
| COMET-DynGen (Bosselut et al., 2021) | ATOMIC | - | - | - | 50.1 | - | - |
| SMLM (Banerjee and Baral, 2020) | * | 65.3 | 38.8 | - | 48.5 | - | - |
| MICO (Su et al., 2022) | ATOMIC | - | 44.2 | - | 56.0 | - | - |
| STL-Adapter (Kim et al., 2022) | ATOMIC | 71.3 | 66.5 | 71.1 | 64.4 | 60.3 | 66.7 |
| **Backbone: RoBERTa-Large** *340M* | | | | | | | |
| RoBERTa-L (MR) (Ma et al., 2021) | ATM-10X | 70.8 | 59.4 | 72.1 | 58.5 | 58.3 | 63.8 |
| △ RoBERTa-L (MR) (Ma et al., 2021) | ATOMIC | 70.8 | 64.2 | 72.1 | 63.1 | 59.2 | 65.9 |
| ◇ **CAR-RoBERTa-L (Ours)** | ATOMIC | $72.3_{\uparrow 1.5}$ | $64.8_{\uparrow 0.6}$ | $73.2_{\uparrow 1.1}$ | $64.8_{\uparrow 1.7}$ | $61.3_{\uparrow 2.1}$ | $67.3_{\uparrow 1.4}$ |
| ◇ **CAR-RoBERTa-L (Ours)** | $ATM^C$ | $72.7_{\uparrow 1.9}$ | $66.3_{\uparrow 2.1}$ | $73.2_{\uparrow 1.1}$ | $64.0_{\uparrow 0.9}$ | $62.0_{\uparrow 2.8}$ | $67.6_{\uparrow 1.7}$ |
| **Backbone: DeBERTa-v3-Large** *435M* | | | | | | | |
| DeBERTa-v3-L (MR) (Ma et al., 2021) | ATM-10X | 75.1 | 71.6 | 79.0 | 59.7 | 71.7 | 71.4 |
| △ DeBERTa-v3-L (MR) (Ma et al., 2021) | ATOMIC | 76.0 | 67.0 | 78.0 | 62.1 | 76.0 | 71.8 |
| ◇ **CAR-DeBERTa-v3-L (Ours)** | ATOMIC | $\underline{78.9}_{\uparrow 2.9}$ | $67.2_{\uparrow 0.2}$ | $\mathbf{78.6}_{\uparrow 0.6}$ | $63.8_{\uparrow 1.7}$ | $\underline{78.1}_{\uparrow 2.1}$ | $\underline{73.3}_{\uparrow 1.5}$ |
| ◇ **CAR-DeBERTa-v3-L (Ours)** | $ATM^C$ | $\mathbf{79.6}_{\uparrow 3.6}$ | $\underline{69.3}_{\uparrow 2.3}$ | $\mathbf{78.6}_{\uparrow 0.6}$ | $64.0_{\uparrow 1.9}$ | $\mathbf{78.2}_{\uparrow 2.2}$ | $\mathbf{73.9}_{\uparrow 2.1}$ |
| **Large Language Models** | | | | | | | |
| GPT-3.5 (text-davinci-003) | - | 61.8 | 68.9 | 67.8 | 68.0 | 60.7 | 65.4 |
| ChatGPT (gpt-3.5-turbo) | - | 69.3 | **74.5** | 75.1 | **69.5** | 62.8 | 70.2 |
| **Supervised Learning & Human Performance** | | | | | | | |
| RoBERTa-L (Supervised) | - | 85.6 | 78.5 | 79.2 | 76.6 | 79.3 | 79.8 |
| DeBERTa-v3-L (Supervised) | - | 89.0 | 82.1 | 84.5 | 80.1 | 84.1 | 84.0 |
| Human Performance | - | 91.4 | 88.9 | 94.9 | 86.9 | 94.1 | 91.2 |

Table 1: Zero-shot evaluation results (%) on five commonsense question answering benchmarks. The best results are **bold-faced**, and the second-best ones are underlined. ↑ indicates the performance gain of our framework (marked with ◇) compared with the results acquired by Ma et al. (2021) on ATOMIC (marked with △). $ATM^C$ stands for the ATOMIC with abstract commonsense knowledge injected. ATM-10X stands for using ATOMIC-10X (West et al., 2022) as the source CSKB $D$. All baseline results are consistent with their original papers.

the LLM directly in a zero-shot setting, where no in-context learning (Min et al., 2022) or chain-of-thought reasoning (Wei et al., 2022) are applied. For every QA entry, the LLM is presented with a question, several choices, and a natural language command that asks it to choose the index of the correct answer directly (Robinson et al., 2022). We then parse the generated outputs to obtain the "predictions" of LLM by using meticulously designed rules and compare them with the ground-truth labels. More details of the baselines and LLM setups can be found in Appendix B.2 and B.3.

**Implementation Details** We use accuracy as the evaluation metric and compare our framework with the following baseline methods. For conceptualization, we leverage an off-the-shelf conceptualizer from Wang et al. (2023a), which is a semi-supervised conceptualization discriminator fine-tuned on labeled conceptualization data from AbstractATOMIC and unlabeled data from ATOMIC. We use a plausibility score $T = 0.9$ to filter out plausible conceptualizations, which results in 440K conceptualization-aided synthetic

QA pairs for training. We employ an AdamW optimizer (Loshchilov and Hutter, 2019) with a learning rate of 7e-6 and a max sequence length of 128 to accommodate QA pairs with different lengths. We select the best checkpoint according to the highest accuracy achieved on the synthetic validation QA set. Each experiment is repeated using three different random seeds, and the average performance is reported. The model is warmed up with 5% of total iterations and evaluated every 1000 global steps, while the margin $\eta$ for the marginal ranking loss is set to 1, in line with the choices made by Ma et al. (2021) and Kim et al. (2022). More details about implementations can be found in Appendix B.4,

## 5.2 Results

The main results are reported in Table 1. For the baselines, DeBERTa-v3-Large (MR) trained on ATOMIC achieves the best performance, followed by ChatGPT. Both achieve an accuracy of more than 70% on average. Our best system, based on DeBERTa-v3-Large and trained on our conceptualization-augmented ATOMIC, achieves

| Augmentation | Div↑ | Exp.Div↑ | Plau.↑ | %F.Neg.↓ | aNLI | CSQA | PIQA | SIQA | WG |
|---|---|---|---|---|---|---|---|---|---|
| N/A *(Baseline)* | N/A | N/A | 88.0 | 45.7 | 76.0 | 67.0 | 78.0 | 62.1 | 76.0 |
| EDA (Wei and Zou, 2019) | 8.10 | 4.67 | 9.33 | 33.0 | 76.5 | 65.6 | 76.6 | 61.4 | 74.9 |
| Word2Vec (Wang and Yang, 2015) | 11.8 | 4.00 | 9.00 | 55.0 | 74.3 | 65.8 | 75.1 | 62.9 | 74.7 |
| GLOVE (Wang and Yang, 2015) | 8.21 | 6.67 | 4.67 | 44.3 | 74.7 | 64.2 | 74.6 | 61.1 | 74.4 |
| BERT-base (Kobayashi, 2018) | 0.81 | 8.33 | 14.3 | 41.7 | 70.4 | 63.9 | 72.4 | 63.5 | 61.0 |
| Synonym (Niu and Bansal, 2018) | 6.92 | 11.0 | 5.67 | 45.0 | 75.5 | 64.9 | 74.5 | 62.5 | 75.7 |
| GPT3-distil (West et al., 2022) | 35.6 | 24.3 | **95.7** | 42.7 | 75.4 | **71.8** | 75.6 | 63.4 | 76.0 |
| **Conceptualization (Ours)** | **48.5** | **37.0** | 90.0 | **22.7** | **79.6** | 69.3 | **78.6** | **64.0** | **78.2** |

Table 2: Comparison results (%) of different augmentation methods against conceptualization. N/A stands for not using any augmentation. Plau. is the expert-evaluated ratio of plausible augmented knowledge, %F.Neg. represents the expert-annotated proportion of false negative options. Div. and Exp.Div. are diversities measured by embedding similarity and expert annotated knowledge coverage. Performances on the right refer to accuracies achieved by the QA model trained on data augmented by each method. The best performances are **bold-faced**.

state-of-the-art results and significantly outperforms all PLM-based baselines on every benchmark, and can advance the average accuracy by 2.1% compared with the same baseline model. It also significantly surpasses the performance of the same model that is trained on ATOMIC-10X with only 10% amount of data (more explanations and experiments in Appendix B.5). Notably, compared with LLMs, our system champions three benchmarks and performs better on average with a 3.7% leap. This indicates that supervision signals from CSKBs are important for downstream applications, and CSKBs aided by conceptualization can significantly enhance this process. Moreover, as an ablation, we study the role of concept-level distractor sampling by discarding conceptualization augmentation and only training the models on ATOMIC, synthesized to QA format with our proposed constraint technique. Comparing the results in Table 1, it can be observed that the concept-level distractor sampling improves the average performance by approximately 1.5%. This demonstrates that our proposed technique is effective, and generating distractors with a stronger positive knowledge recall is helpful in synthesizing QA pairs that are both fair and informative.

## 6 Analysis and Discussion

In this section, we study the effects of conceptualization and the reasons contributing to CAR's success. First, we conduct expert evaluations on the synthetic QA pairs to study the quality and diversity of different CSKB augmentation methods in comparison with conceptualization. Second, we conduct training dynamics (Swayamdipta et al., 2020) analysis to show that conceptualization-aided QA pairs can provide more *ambiguous* examples help-

ful for training. Finally, we study the impact of filtering ATOMIC10X with different critic thresholds, the ablations of CAR, and the effect of conceptualization from an out-of-domain generalization perspective in Appendix B.5, B.7, and B.8.

### 6.1 Comparisons With Data Augmentations

To demonstrate the effectiveness of our proposed conceptualization method, we conduct comprehensive analyses with other data augmentations that expand the semantic coverage of CSKBs in a similar way as conceptualization using both expert and automatic evaluations. We use EDA, augmenting with word embedding (Word2Vec; Mikolov et al., 2013 and GLOVE; Pennington et al., 2014), contextual embedding (BERT; Devlin et al., 2019), and synonym (WordNet; Miller, 1995) as baselines. To align all the baselines for fair comparisons, we only augment the identified instance $i \in h$ in each ATOMIC triple's head event $h$ according to the number of its valid conceptualizations $|C_h|$. Additionally, we randomly sample another same amount of knowledge from ATOMIC-10X into ATOMIC as a form of augmentation by distilling GPT3 (Brown et al., 2020) to set distilling an LLM as another baseline (more explanations in Appendix B.5).

We comprehensively analyze the comparison using three dimensions: diversity, quality of synthetic QA pairs, and zero-shot commonsense QA performances. Three expert annotators are recruited to facilitate our evaluations who are undergraduate or graduate students actively involved in commonsense research. They demonstrate a high level of agreement among themselves, with an IAA of 83% in terms of pairwise agreement and a Fleiss Kappa score (McHugh, 2012) of 0.64, comparable to 0.62, as reported by (Ma et al., 2021).

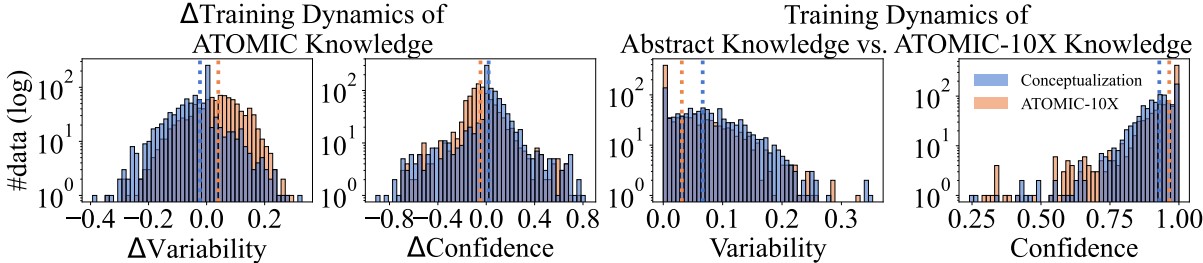

Figure 4: Analyses on training dynamics of different knowledge. The dotted lines refer to the median values.

**Diversity.** First, we study whether augmentations can introduce new knowledge to the training set. We begin by calculating the average cosine similarity of each ATOMIC triple and its augmented siblings from each method according to their SentenceBERT (Reimers and Gurevych, 2019) embeddings. For ATOMIC-10X, we regard the sampled knowledge as augmentations. The complement of average similarity across all ATOMIC triples serves as an automatically measured diversity (Div.). Meanwhile, we retrieve top-10 similar triples from ATOMIC for each augmented triple according to their SentenceBERT similarity. The experts are asked to annotate whether each triple can be semantically covered by their retrievals. We define the expert-evaluated diversity as the ratio of uncovered triples among 300 samples. Table 2 shows that conceptualization champions both metrics, indicating that the introduced abstract knowledge is diverse and lacking in existing CSKBs, which is helpful in expanding their knowledge coverage.

**Quality of Synthetic QA Pairs.** Next, we synthesize the augmented triples into QA pairs with their head events' keywords and augmentations as constraints. We then sample 300 QA pairs for each method and ask the same experts to perform expert evaluations by annotating the correctness of each QA pair's ground-truth answer and whether the distractors could also be plausible with respect to the augmented head event. This evaluates the plausibility ratio of the augmented knowledge and the ratio of QA pairs containing false negative distractors. Table 2 shows that the majority of augmented knowledge is implausible, and they fail to enhance distractors sampling. Conceptualization, on the other hand, maintains being highly plausible and can effectively eliminate false negative distractors. Expert annotators also achieve a remarkable accuracy of 86% while working on 300 randomly sampled question-answer pairs, surpassing the 80%

accuracy reported by Ma et al. (2021).

**Zero-shot Commonsense QA Performances.** Finally, we train DeBERTa-v3-Large models on the QA pairs synthesized from the concatenation of both original and augmented ATOMIC triples from each method. Only keywords of each head event are used as their constraints. The models are trained using a marginal ranking loss, as explained in Section 4.3, and evaluated on five QA benchmarks in a zero-shot manner. Performances by different methods are shown in Table 2. We observe that conceptualization outperforms all other augmentations on average and successfully improves the model's zero-shot commonsense reasoning ability.

**Comparison with ATOMIC-10X.** Augmenting ATOMIC10X appears to be a promising option as it contains a wealth of valuable commonsense knowledge. However, despite its diverse and high-quality knowledge, Table 2 demonstrates that the model cannot leverage this information effectively. One possible explanation is that the model's performance is hindered by the significantly high number of false-negative distractors. This issue arises because the knowledge distilled from GPT-3 tends to be versatile, resulting in many tail events being general and vague. These events can be applied to a large collection of heads, which leads to false negative options. More experiments and case studies are in Appendix B.5 and C, respectively.

### 6.2 Training Dynamics Analysis

Training dynamics effectively assess a model's confidence and variability for individual instances when training on a large dataset. In the context of QA, we define confidence as the model's certainty when assigning the correct label to the ground-truth option compared to distractors, as indicated by the logit difference. Variability, on the other hand, refers to the fluctuation of confidence over time. These insights can aid in analyzing the model's

| Model | CSKB | Avg. |
|---|---|---|
| RoBERTa-L (MR) (Ma et al., 2021) | CWWV | 64.8 |
| MTL (Kim et al., 2022) | CWWV | 63.7 |
| ZS-Fusion (Kim et al., 2022) | CWWV | 64.7 |
| CAR-RoBERTa-L (Ours) | CWWV$^C$ | **65.8** |

Table 3: Experiments on the generalizability of CAR on other CSKBs (CWWV).

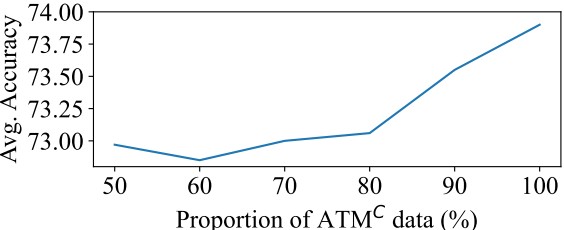

Figure 5: Average accuracy achieved by models trained on our training set downsampled to several ratios.

behavior when different knowledge is introduced into the training set. More explanations are in Appendix B.6.

In this section, we examine the impact of abstract commonsense knowledge (conceptualization) and GPT3-distilled knowledge (ATOMIC-10X) by exploring their training dynamics on two sets of data. We train three QA models on synthetic QA pairs from conceptualization-augmented ATOMIC, ATOMIC10X-augmented ATOMIC, and the original ATOMIC, which serves as the baseline. First, we randomly select the same 1,000 QA pairs synthesized from the original ATOMIC and calculate their training dynamics using these three models. The left side of Figure 4 displays the alterations caused by the two augmentation methods in comparison with the baseline. It is evident that introducing abstract commonsense knowledge through conceptualization significantly reduces the model's average variability and enhances its confidence in learning the knowledge from the original ATOMIC. In contrast, incorporating knowledge from ATOMIC-10X produces the opposite effect.

Second, we check the training dynamics on 1,000 randomly sampled QA pairs synthesized from abstract commonsense knowledge and another 1,000 from knowledge in ATOMIC-10X. The rightmost plots in Figure 4 reveal that, compared to ATOMIC-10X, conceptualization introduces knowledge with higher variability and lower confidence, making it more *ambiguous* and challenging for the model to learn. As Swayamdipta et al. (2020) suggest, such data contributes to a more robust model to out-of-distribution (OOD) data, which are downstream QA datasets in our case. Therefore, we conclude that conceptualization is superior to ATOMIC-10X as abstract knowledge, on the one hand, makes the original knowledge more easy-to-learn to aid optimization, and on the other hand, provides more *ambiguous* examples to boost OOD generalization.

### 6.3 Impact of Training Data Size

In Figure 5, we present the influence of the number of training examples against the final performance, which reveals a clear and intuitive trend of a positive correlation between the amount of training data and overall performance.

### 6.4 Generalization to other CSKBs

We explore the feasibility of transferring our framework to CSKBs other than ATOMIC. We take the CWWV dataset as an example, which comprises multiple CSKBs, including ConceptNet (Speer et al., 2017), WordNet (Miller, 1995), and Wiki-Data (Vrandecic and Krötzsch, 2014). We use the off-the-shelf GPT2 conceptualizer (Wang et al., 2023a) and ChatGPT as two flexible generative conceptualizers. The generated conceptualizations are then transformed into abstract knowledge and integrated into the CWWV dataset. The experimental results are presented in Table 3, which shows an improvement of over 1% compared to all baselines leveraging CWWV as the source of knowledge, indicating CAR's generalizability to other CSKBs. More details are presented in the Appendix B.9.

### 7 Conclusions

In this paper, we present CAR, a pioneering framework for zero-shot commonsense QA empowered by conceptualization. Our approach surpasses even large language models on five QA benchmarks, achieving state-of-the-art performance on average. Our analyses reveal that conceptualization can improve the sampling of negative examples, and abstract knowledge is more helpful compared with those distilled from GPT3 as it provides more ambiguous knowledge to support OOD generalization. These findings demonstrate the substantial benefits of introducing conceptualization and abstract knowledge into zero-shot commonsense reasoning.

## Limitations

One limitation of this paper is that the proposed CAR framework has only been validated on the ATOMIC dataset. While previous works (Ma et al., 2021; Kim et al., 2022; Dou and Peng, 2022) have studied the zero-shot commonsense question answering task by consolidating multiple CSKBs, including ATOMIC (Sap et al., 2019a), ConceptNet (Speer et al., 2017), WordNet (Miller, 1995), VisualGenome (Krishna et al., 2017), and WikiData (Vrandecic and Krötzsch, 2014), our work only utilizes ATOMIC (more details discussed in Appendix B.2). This was mainly due to the availability of conceptualizations for the CSKB, with only AbstractATOMIC (He et al., 2022) being available as the conceptualized expansion of ATOMIC, while other CSKBs lack such resources. Additionally, ATOMIC has been shown to play the most critical role in experimental results by Ma et al. (2021). Nonetheless, such limitation does not restrict CAR's potential to seek further improvements from incorporating other CSKBs, as conceptualization frameworks, such as CAT (Wang et al., 2023a), can be applied to other CSKBs and provide the required resources for CAR to operate. Thus, we believe CAR can overcome such limitations and still possess the potential to improve with more CSKB-associated conceptualization resources available.

## Ethics Statement

This paper presents CAR, a novel framework for zero-shot commonsense question answering that achieves state-of-the-art performance via *conceptualization*. All datasets used, including ATOMIC, AbstractATOMIC, and commonsense question-answering benchmarks, are publicly available and shared via open-access licenses solely for research purposes, consistent with their intended usage. These datasets are anonymized and desensitized, ensuring that no data privacy issues are involved. Moreover, the CAR framework is a question-answering system that selects the most plausible choice from a list of options and does not yield any private, offensive, biased, or sensitive information or social and political issues. The expert annotations are performed by the authors of this paper as part of their contribution, who are graduate and undergraduate students working on machine commonsense in natural language processing, and they are fully aware of the annotation protocol and the intended use of their annotations. They are well-trained with specially designed instructions and have voluntarily agreed to participate without compensation. Based on this, the authors believe that this paper does not raise any ethical concerns to the best of their knowledge.

## Acknowledgements

The authors would like to thank Haochen Shi for his help in implementing the training dynamics and the anonymous reviewers for their constructive comments. The authors of this paper were supported by the NSFC Fund (U20B2053) from the NSFC of China, the RIF (R6020-19 and R6021-20), and the GRF (16211520 and 16205322) from RGC of Hong Kong. We also thank the support from the UGC Research Matching Grants (RMGS20EG01-D, RMGS20CR11, RMGS20CR12, RMGS20EG19, RMGS20EG21, RMGS23CR05, RMGS23EG08). We also gratefully acknowledge the support of the Swiss National Science Foundation (No. 215390), Innosuisse (PFFS-21-29), the EPFL Science Seed Fund, the EPFL Center for Imaging, Sony Group Corporation, and the Allen Institute for AI.

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

# Appendices

## A  Benchmark Descriptions

In this section, we introduce more details regarding five evaluation benchmarks.

**Abductive NLI (aNLI)** (Bhagavatula et al., 2020) is a Natural Langauge Inference (NLI) benchmark that aims to infer the most plausible explanation based on a given causal situation. For each question sample, the model is required to choose the more plausible hypothesis out of two options that fit the beginning and end of a story. This benchmark evaluates the model's abductive reasoning ability, which typically requires commonsense reasoning.

**CommonsenseQA (CSQA)** (Talmor et al., 2019) is a question-answering benchmark that evaluates a broad range of commonsense aspects. Each sample contains a question and five choices. The question and some choices are generated from subgraphs of ConceptNet (Speer et al., 2017) while crowdsourcing annotators also annotate some distractors. This benchmark evaluates the model's concept-level commonsense reasoning ability.

**PhysicalIQA (PIQA)** (Bisk et al., 2020) is a question-answering benchmark that requires the model to select the more plausible option out of two possible continuations given a common scenario that requires physical commonsense to infer. This benchmark evaluates the model's physical commonsense reasoning ability.

**SocialIQA (SIQA)** (Sap et al., 2019b) is a question-answering benchmark that requires reasoning about social interactions. Each sample contains a context that is derived from ATOMIC (Sap et al., 2019a), a question, and three choices. The questions are automatically generated using nine templates that correspond to the nine relations in ATOMIC, and the correct answers are crowdsourced. This benchmark evaluates the model's reasoning ability for emotional and social commonsense in daily situations.

**WinoGrande (WG)** (Sakaguchi et al., 2021) is a pronoun resolution benchmark. Each sample contains an emphasized pronoun and a short context description. The model is asked to choose the correct reference given two options. This benchmark evaluates the model's pronoun resolution ability, which is also part of commonsense knowledge.

In our experiments, we use the validation splits of these benchmarks as the official testing sets may

|  | aNLI | CSQA | PIQA | SIQA | WG |
|---|---|---|---|---|---|
| #QA Pairs | 1,532 | 1,221 | 1,838 | 1,954 | 1,267 |
| #Options | 2 | 5 | 2 | 3 | 2 |

Table 4: Statistics on the number of QA pairs and the number of options for each question within each benchmark's validation split.

not be publicly available. Detailed statistics on the number of QA pairs and the number of options per question are reported in Table 4.

## B  Additional Explanations and Analyses

In this section, we aim to cover additional details regarding the CSKB conceptualization in CAR (Appendix B.1), implementations of our system (Appendix B.4), baselines (Appendix B.2 and B.3), experiments using ATOMIC-10X (Appendix B.5), analyses (Appendix B.6, B.7, and B.8), and generalizability experiments (Appendix B.9) that are not covered in the body text due to space constraints.

### B.1  Definitions and Statistics of CSKB Conceptualization

Conceptualization plays a crucial role in generalizable commonsense reasoning. Previous studies have demonstrated its potential in aiding commonsense inference modeling (Wang et al., 2023a) and commonsense knowledge graph construction (Yu et al., 2023; Zhang et al., 2022). In our paper, we follow the definition of conceptualization proposed by He et al. (2022) and Wang et al. (2023a) in conceptualizing an instance within an event to a concept: **(1) Events**: Each event represents a commonly observed occurrence that encompasses valuable subsequential or inferential commonsense knowledge. In AbstractATOMIC, the events are the head events of all triples in ATOMIC without a wildcard ('_'). **(2) Instances**: Within each event, multiple instances have been identified with semantic parsing tools, representing specific components of the event that are worthy of conceptualization. **(3) Concepts**: Concepts are the conceptualization of each instance. These concepts are thus extracted from Probase/WordNet and further validated by human annotators or critic filtering models.

For an event $e$, which is the head of an ATOMIC triple, an instance refers to either an entity within the event or the complete event itself. Multiple instances can exist within a single event, denoted as $i_1, i_2, i_3, \ldots, i_n \in e$. A concept corre-

sponds to the conceptualization of an instance, and multiple conceptualizations can be associated with a single instance, as demonstrated by $(i_1, c_1, 1), (i_1, c_1, 2), (i_1, c_1, 3), ..., (i_2, c_2, 1), ...,$ $(i_n, c_n, 1), ..., (i_n, c_n, m)$. For instance, consider the event "PersonX is drunk when exiting the bar." In this case, two instances can be identified: "PersonX is drunk when exiting the bar" and "bar." The conceptualization for the instance "PersonX is drunk when exiting the bar" may include "drunk" or "enjoyed," while the instance "bar" can be conceptualized as an "entertainment place" or a "fun place."

In this paper, we leverage the AbstractATOMIC dataset, provided by He et al. (2022), as our primary source for conceptualizations. AbstractATOMIC is a benchmark for conceptualized commonsense knowledge that is built upon the ATOMIC dataset (Sap et al., 2019a). It contains three folds of data, each conditioned on the original commonsense knowledge triples $(h, r, t)$ in ATOMIC.

In the first fold, He et al. (2022) identify all possible instances $\{i_1, i_2, i_3, \cdots | i \subseteq h\}$ in each ATOMIC head event, using syntactic parsing through a spaCy[1] parser and matching with five human-defined rules. It is important to note that, unlike traditional entity-level conceptualization benchmarks, the identified instance in AbstractATOMIC can also be the entire head event $i = h$.

In the next fold, each identified instance $i$ is heuristically matched against Probase (Wu et al., 2012) and WordNet (Miller, 1995) via Gloss-BERT (Huang et al., 2019) to find their corresponding conceptualization candidates. Human annotations are conducted to verify part of the plausibility of the conceptualization candidates. To pseudo-label unannotated conceptualizations, we use a semi-supervised conceptualization discriminator provided by Wang et al. (2023a) and set a threshold of $T = 0.9$ to filter out plausible conceptualizations. Additionally, we utilize a GPT2-based (Radford et al., 2019) generator, trained on the concatenation of annotated and positively pseudo-labeled conceptualizations, to generate additional conceptualizations for further expanding the size of the conceptualization bank.

However, it is worth noting that such conceptualization may not yield plausible abstract knowledge when $(r, t)$ is connected back to $h_c$, where $h_c$ is

[1] https://spacy.io/

| | $D_h^l$ | $D_h^u$ | Total |
|---|---|---|---|
| #Unq. event | 7,196 | 15,165 | 15,388 |
| #Unq. instance | 7,935 | 20,843 | 21,493 |
| #Unq. concept | 20,036 | 20,367 | 31,227 |
| Avg. #concept/event | 18.21 | 24.57 | 32.73 |
| Avg. #concept/instance | 16.51 | 17.88 | 23.43 |

Table 5: Statistics of conceptualizations used in CAR, as reported by Wang et al. (2023a). $D_h^l$ stands for human-annotated conceptualizations and $D_h^u$ are unlabeled conceptualizations. Unq stands for unique, and Avg refers to average.

obtained by replacing $i \in h$ with its conceptualizations. This is because the process of conceptualizing a head event omits its context in $(r, t)$. Thus, the last fold of data stores the plausibility of such abstract commonsense triples $(h_c, r, t)$, where human annotations are conducted to verify part of the triples' plausibilities. In addition, we adopt a semi-supervised instantiation discriminator, provided by Wang et al. (2023a), to pseudo-label the unannotated triples. Another threshold, $T = 0.9$, is set to filter out plausible abstract triples.

In the CAR framework, for every ATOMIC event $h$, we retrieve every instance $i$'s plausible conceptualizations $\{c_{i,1}, c_{i,2}, \cdots\}$ from all plausible conceptualizations derived in the second fold to serve as the distractor sampling constraint. We also augment the original $(h, r, t)$ triples with their plausible $(h_c, r, t)$ siblings from both human-annotated and pseudo-labeled triples, as explained in the last fold. These knowledge triples are then synthesized into QA pairs using our proposed method to train the model to perform general reasoning. Detailed statistics for the conceptualizations and abstract commonsense triples we finally obtained from the AbstractATOMIC dataset are reported in Table 5 and Table 6, respectively.

### B.2 Baseline Performances

For SMLM (Banerjee and Baral, 2020), we adopt the official implementation of Banerjee and Baral (2020), which employs the CSKB that exhibits the highest alignment with each task. Specifically, SocialIQA uses ATOMIC, while CommonsenseQA uses ConceptNet. For STL-Adapter (Kim et al., 2022), only those trained on ATOMIC are used for comparison in the body text. In this paper, all baseline performances are solely based on their officially reported results in their respective papers.

As noted in the Limitations section, previous re-

| Relation | ATOMIC | $D_t^l$ | $D_t^u$ |
|---|---|---|---|
| xEffect | 78,832 | 12,168 | 412,455 |
| oEffect | 28,351 | 3,526 | 113,301 |
| xWant | 101,249 | 15,312 | 177,745 |
| oWant | 43,079 | 5,408 | 38,938 |
| xReact | 62,969 | 8,923 | 295,044 |
| oReact | 26,570 | 3,030 | 104,038 |
| xNeed | 74,272 | 11,733 | 378,442 |
| xAttr | 110,791 | 14,249 | 275,224 |
| xIntent | 45,490 | 6,848 | 234,948 |
| Total | 572,053 | 81,197 | 2,030,135 |

Table 6: Statistics of abstract commonsense triples used in CAR, as reported by Wang et al. (2023a). $D_t^l$ stands for human-annotated triples and $D_t^u$ are unlabeled triples.

search in this area has primarily focused on using four CSKBs, namely ATOMIC (Sap et al., 2019a), ConceptNet (Speer et al., 2017), WordNet (Miller, 1995), and WikiData (Vrandecic and Krötzsch, 2014). In order to comprehensively benchmark our framework's performance in the field of zero-shot commonsense QA, we compare our results on ATOMIC against baseline methods that use multiple CSKBs despite the unbalanced amount of knowledge in such a comparison. Table 11 presents a full comparison of our method with all existing baselines. Notably, for models based on RoBERTa-Large, our approach trained only on abstract knowledge injected ATOMIC achieves second place in the leaderboard, falling only behind Kim et al. (2022) with four CSKBs. While this comparison may be unfair due to the unbalanced amount of knowledge, it provides a strong justification for the excellent performance of our system. Our DeBERTa-v3-Large-based model still surpasses all baselines on average, indicating the necessity of leveraging a strong pre-trained language model.

## B.3 Benchmarking Large Language Models

We then discuss our method for benchmarking large language models on five commonsense QA benchmarks. The emergence of Large Language Models (LLMs), such as ChatGPT (OpenAI, 2022), has been the hot trend in recent NLP research. Numerous studies have evaluated the capability of LLMs on various NLP downstream tasks. Among them, Qin et al. (2023); Chan et al. (2023) have shown that ChatGPT can achieve competitive performance on commonsense reasoning tasks, such as CommonsenseQA (Talmor et al., 2019), Wino-

Grande (Sakaguchi et al., 2021), and Commonsense Knowledge Base Population (Fang et al., 2021b,a). In this study, we aim to benchmark ChatGPT's zero-shot performance on five QA evaluation benchmarks used in our zero-shot commonsense QA task. Following (Robinson et al., 2022), we design and leverage a batch of prompts, as shown in Table 7, to probe ChatGPT's predictions. The prompt provides ChatGPT with a question and its possible choices, along with a natural language command to control the response action of ChatGPT. We then parse the generations using our meticulously designed rules, where punctuations and irrelevant wordings will be dropped, and the first choice-letter prediction will be identified as ChatGPT's answer. Specifically, if ChatGPT hesitates and cannot make a concrete prediction, it will be counted as a wrong answer. The benchmarking results are shown in Table 1. We observe that ChatGPT demonstrates superior performance compared to GPT3.5 (Ouyang et al., 2022) and excels in tasks such as CommonsenseQA (Talmor et al., 2019) and SocialIQA (Sap et al., 2019b), potentially due to the high frequency of their questions and answers in large text corpora. However, its performance on the remaining three benchmarks is suboptimal, suggesting that they are more challenging and require more complex reasoning (Bai et al., 2023; Ding et al., 2023) and implicit commonsense knowledge to solve. This intriguing outcome warrants further investigation to determine the reasons behind it and explore methods to boost the LLM's abilities in these challenging benchmarks.

Generally speaking, CAR and conceptualization own the advantage over the large language model in the following aspects: (1) Smaller Model Size: The CAR framework offers models that are significantly smaller in scale compared to LLMs (0.2% of a standard 175 billion parameter GPT-3 model) while maintaining comparable performance in a zero-shot setting. Such size makes it more efficient in terms of training and deployment. In contrast, advanced prompting techniques used in LLMs require extensive computational resources, making the conceptualization-based model more versatile and accessible to researchers with limited access or resources for deploying LLMs. (2) Broader Commonsense Knowledge: Conceptualization provides a broader range of commonsense knowledge compared to current CSKBs. Integrating this type of knowledge into generative models has been shown

| Task | Prompt | Gen |
|------|--------|-----|
| aNLI | Premise: Jim decided to be a rockstar.
Choice A: but didn't know how to play an instrument. Jim signed up for guitar lessons.
Choice B: Jim knew he would need to have a nickname. Jim signed up for guitar lessons.
Which one is more likely to happen, given the premise? Only answer A or B without any other word. | A. |
| CSQA | Question: He was at the gym trying to build muscle, what is it called that he is trying to build muscle on?
Choice A: body of animal
Choice B: arm
Choice C: bodybuilder
Choice D: body of dog
Choice E: human body
Which choice is correct? Only answer A or B or C or D or E without any other word. | C |
| PIQA | Goal: To remove an avocado from the shell
Choice A: cut the avocado lengthwise, remove the pit, and scoop with a spoon
Choice B: cut the avocado width wise, remove the pit, and scoop with a spoon
Which choice can achieve the goal? Only answer A or B without any other word. | A. |
| SIQA | Question: Robin went to the polls and posted her ballot for the candidate she wanted.
As a result, Robin wanted to:
Choice A: bomb the candidate
Choice B: attend a rally
Choice C: go home.
Which choice is correct? Only answer A or B or C without any other word. | C. |
| WG | Question: Jessica enjoyed a simple, basic life with Betty, but
Choice A: Jessica was bored having a quiet existence.
Choice B: Betty was bored having a quiet existence.
Which choice is correct? Only answer A or B without any other word. | A |

Table 7: Prompts used for evaluating GPT3.5 and ChatGPT. Gen. refers to the generated outputs from ChatGPT.

to enhance their performance in commonsense reasoning tasks (Wang et al., 2023a). Such knowledge can also be dynamically encoded in language models during inference time (Chen et al., 2023b). (3) Advanced Prompting of LLMs: Conceptualization data introduces the potential for a more advanced prompting framework of reasoning with LLMs. The process of conceptualization and the instantiation of knowledge play a crucial role in reasoning. Thus, future work may consider introducing the "chain of concept" reasoning process to further advance the popular "chain of thought" reasoning paradigm (Wei et al., 2022; Wang et al., 2023b).

### B.4 Implementation Details

We present additional implementation details for building our system. For the pre-trained language models, We use PLMs from the Huggingface Transformers[2] Library (Wolf et al., 2020) as the vanilla model checkpoints. Our system relies heavily on the open-source code repository[3] provided by Ma et al. (2021) for synthesizing QA pairs and training the QA models. To optimize the model, we employ an AdamW optimizer (Loshchilov and Hutter,

2019) with a learning rate of 7e-6 and a max sequence length of 128 to accommodate QA pairs with different lengths. When evaluating the model on downstream commonsense QA benchmarks, a maximum sequence length of 80 is used. We select the best checkpoint according to the highest accuracy achieved on the synthetic validation QA set. Each experiment is repeated using different random seeds three times, and the average performance is reported. To overcome the limited GPU memory issue, we utilize gradient accumulation with a gradient accumulated four steps before every descent, and each step calculates the gradient of eight data samples. The model is warmed up with 5% of total iterations and evaluated every 1000 global steps, while the margin $\eta$ for the marginal ranking loss is set to 1, in line with the choices made by Ma et al. (2021) and Kim et al. (2022). The Huggingface model code for our RoBERTa-Large model is `roberta-large`, and the one for our DeBERTa-v3-Large model is `microsoft/deberta-v3-large`. All of our experiments are conducted on eight NVIDIA A100 GPUs, each with 40G graphical memory. Training a RoBERTa-based model typically requires 14G graphical memory, while training DeBERTa-based models requires 30G graphical memory.

---

[2]https://huggingface.co/docs/transformers/
[3]https://github.com/Mayer123/HyKAS-CSKG

| Model | CSKB | Critic | a-NLI | CSQA | PIQA | SIQA | WG | Avg. |
|---|---|---|---|---|---|---|---|---|
| **Backbone: RoBERTa-Large** 340M | | | | | | | | |
| RoBERTa-L (MR) | ATOMIC | N/A | 70.8 | 64.2 | 72.1 | 63.1 | 59.2 | 65.9 |
| RoBERTa-L (MR) | ATM-10X | 0.9 | 69.6 | 58.1 | 72.3 | 58.3 | 57.2 | 63.1 |
| RoBERTa-L (MR) | ATM-10X | 0.8 | 70.1 | 58.9 | 71.5 | 58.2 | 57.7 | 63.3 |
| RoBERTa-L (MR) | ATM-10X | 0.7 | 70.8 | 59.4 | 72.1 | 58.5 | 58.3 | 63.8 |
| RoBERTa-L (MR) | ATM-10X | 0.5 | 68.7 | 56.8 | 71.7 | 58.4 | 60.1 | 63.1 |
| RoBERTa-L (MR) | ATM-10X | 0.0 | 70.7 | 58.3 | 71.7 | 58.2 | 57.5 | 63.3 |
| RoBERTa-L (MR) | ATM$^{\text{ATM-10X}}$ | 0.9 | 71.7 | 66.3 | 73.2 | 62.8 | 60.7 | 66.9 |
| RoBERTa-L (MR) | ATM$^{\text{ATM-10X}}$ | 0.8 | 71.8 | 66.0 | 73.2 | 61.7 | 59.5 | 66.4 |
| RoBERTa-L (MR) | ATM$^{\text{ATM-10X}}$ | 0.7 | 71.6 | 65.6 | 72.9 | 62.2 | 59.8 | 66.4 |
| RoBERTa-L (MR) | ATM$^{\text{ATM-10X}}$ | 0.5 | 72.0 | 65.4 | 72.9 | 62.0 | 60.5 | 66.6 |
| RoBERTa-L (MR) | ATM$^{\text{ATM-10X}}$ | 0.0 | 71.6 | 66.3 | 73.3 | 62.9 | 61.0 | 67.0 |
| CAR-RoBERTa-L (Ours) | ATOMIC | N/A | 72.3 | 64.8 | 73.2 | 64.8 | 61.3 | 67.3 |
| CAR-RoBERTa-L (Ours) | ATM$^C$ | N/A | 72.7 | 66.3 | 73.2 | 64.0 | 62.0 | 67.6 |
| **Backbone: DeBERTa-v3-Large** 435M | | | | | | | | |
| DeBERTa-v3-L (MR) | ATOMIC | N/A | 76.0 | 67.0 | 78.0 | 62.1 | 76.0 | 71.8 |
| DeBERTa-v3-L (MR) | ATM-10X | 0.9 | 74.5 | 70.8 | 78.9 | 59.7 | 72.2 | 71.2 |
| DeBERTa-v3-L (MR) | ATM-10X | 0.8 | 74.2 | 70.6 | **79.5** | 59.2 | 70.7 | 70.8 |
| DeBERTa-v3-L (MR) | ATM-10X | 0.7 | 74.6 | 69.9 | 79.3 | 60.0 | 70.2 | 70.8 |
| DeBERTa-v3-L (MR) | ATM-10X | 0.5 | 74.1 | 70.4 | 78.8 | 58.9 | 70.1 | 70.5 |
| DeBERTa-v3-L (MR) | ATM-10X | 0.0 | 75.1 | 71.6 | 79.0 | 59.7 | 71.7 | 71.4 |
| DeBERTa-v3-L (MR) | ATM$^{\text{ATM-10X}}$ | 0.9 | 75.4 | 71.3 | 73.4 | 61.7 | 75.3 | 71.4 |
| DeBERTa-v3-L (MR) | ATM$^{\text{ATM-10X}}$ | 0.8 | 75.4 | 71.8 | 75.6 | 63.4 | 76.0 | 72.4 |
| DeBERTa-v3-L (MR) | ATM$^{\text{ATM-10X}}$ | 0.7 | 74.9 | 71.2 | 77.4 | 61.8 | 76.2 | 72.3 |
| DeBERTa-v3-L (MR) | ATM$^{\text{ATM-10X}}$ | 0.5 | 74.8 | 71.2 | 77.1 | 61.7 | 75.7 | 72.1 |
| DeBERTa-v3-L (MR) | ATM$^{\text{ATM-10X}}$ | 0.0 | 76.2 | 71.0 | 75.8 | 62.8 | 75.8 | 72.3 |
| CAR-DeBERTa-v3-L (Ours) | ATOMIC | N/A | 78.9 | 67.2 | 78.6 | 63.8 | 78.1 | 73.3 |
| CAR-DeBERTa-v3-L (Ours) | ATM$^C$ | N/A | **79.6** | 69.3 | 78.6 | 64.0 | **78.2** | **73.9** |
| **Large Language Models** | | | | | | | | |
| GPT-3.5 (text-davinci-003) | N/A | N/A | 61.8 | 68.9 | 67.8 | 68.0 | 60.7 | 65.4 |
| ChatGPT (gpt-3.5-turbo) | N/A | N/A | 69.3 | **74.5** | 75.1 | **69.5** | 62.8 | 70.2 |

Table 8: Zero-shot evaluation results (%) on five commonsense question answering benchmarks using different critic thresholds for filtering ATOMIC-10X. The best results are **bold-faced**, and the second-best ones are underlined. ATM$^C$ stands for the ATOMIC with abstract commonsense knowledge injected. ATM-10X stands for using ATOMIC-10X (West et al., 2022) as the source CSKB $D$. ATM$^{\text{ATM-10X}}$ indicates the ATOMIC with sampled knowledge from ATOMIC-10X injected. Critic indicates the lower bound for filtering knowledge from ATOMIC-10X, which means that only knowledge with a critic score above the threshold will be selected.

## B.5 Experiments with ATOMIC-10X

ATOMIC-10X is a machine-generated corpus developed by West et al. (2022) using the symbolic knowledge distillation framework. They employed a selective distillation approach to extract knowledge from large language models like GPT-3 (Brown et al., 2020) by prompting them with head events and commonsense relations from the ATOMIC dataset. The extracted knowledge was used to train a student model to generate symbolic knowledge graphs evaluated using a separate critic model. The resulting corpus, ATOMIC-10X, surpassed the human-generated corpus ATOMIC2020 (Hwang et al., 2021) in scale, accuracy, and diversity.

In this section, we provide additional explanations regarding the usage of ATOMIC-10X in our paper and conduct further experiments to explore its impact on zero-shot commonsense QA. Specifically, we study the role of filtering the knowledge in ATOMIC-10X using multiple critic thresholds to acquire high-quality knowledge and improve model performance.

We utilize ATOMIC-10X in two distinct scenarios. First, as discussed in Section 5.2, we directly train our QA models on ATOMIC-10X without integrating other CSKBs, such as ATOMIC and AbstractATOMIC. We source all questions, answers, and distractors from ATOMIC-10X and follow its original train/dev/test partitions to divide the data. The lemmatized tokens of the head event, excluding commonly seen subjects, prepositions, and stopwords, are used as keywords for each piece of knowledge, and the original QA synthesis pipeline proposed by Ma et al. (2021) is applied. To ensure the quality of the knowledge from

ATOMIC-10X, we set multiple critic thresholds to filter the dataset accordingly. The QA models are trained using marginal ranking loss on four subsets of ATOMIC-10X, each with a different critic threshold of 0.9, 0.8, 0.7, and 0.5, along with an additional model trained on the complete ATOMIC-10X dataset. Finally, we evaluate these models on five commonsense QA benchmarks in a zero-shot setting and report the results in Table 8. Specifically, models trained solely on ATOMIC-10X using critic thresholds of 0.7 (RoBERTa) and 0.0 (DeBERTa) for filtering are responsible for the results reported in Table 1. We observe that even using high critic thresholds to filter the knowledge in ATOMIC-10X, the model still fails to improve beyond marginal. Meanwhile, training the models only on ATOMIC-10X fails to surpass training on ATOMIC, which indicates that the amount of knowledge is not the critical element to determining the performance. Rather, it should be the diversity and quality of knowledge, where the human-annotated knowledge from ATOMIC is superior to those machine-generated ones from ATOMIC. Nonetheless, none of the models outperform those trained on conceptualization-augmented ATOMIC using our CAR framework, which further validates the strengths of CAR.

In the second scenario, as discussed in Section 6.1, we utilize ATOMIC-10X as a means of augmentation to extend the original ATOMIC dataset. This is achieved by randomly selecting a specific number of knowledge triples from ATOMIC-10X, equivalent to the total number of plausible abstract commonsense knowledge in AbstractATOMIC, and merging them back into the original dataset. The triples in the resulting ATOMIC10X-augmented ATOMIC are then transformed into QA pairs and used to train our model following the original pipeline suggested by Ma et al. (2021). Similar to the previous scenario, we set four thresholds, namely 0.9, 0.8, 0.7, and 0.5, to filter the triples in ATOMIC-10X for augmentation quality control. In this way, the QA pairs' distractors can come from ATOMIC and ATOMIC-10X. The models are then trained and evaluated on five benchmarks. Their zero-shot commonsense QA evaluation results are reported in Table 8, and the best model, trained using a critic threshold of 0.8 for filtering with DeBERTa-v3-large as the backbone, is responsible for the results indicated in Table 2. Interestingly, we observe that leveraging

the knowledge in ATOMIC-10X, either for direct training or augmentation, occasionally improves the model's performance on a specific benchmark. However, it fails to boost the overall performance across all benchmarks on average, which is considered a closer metric for evaluating the generalizable reasoning ability of a commonsense QA model. Thus, we come to the conclusion that ATOMIC-10X is inconsistently helpful in improving the zero-shot commonsense QA performances, with most times failing to improve, while conceptualization resolves such issues and can benefit the model across all benchmarks significantly. One potential reason is that ATOMIC-10X main contain noise that are not benefitial to the task of zero-shot commonsense QA, as demonstrated by Deng et al. (2023).

## B.6 Training Dynamic Definitions

Training dynamic, as proposed by Swayamdipta et al. (2020), refers to the analysis of a model's behavior on individual instances during training on large datasets. This analysis examines the model's *confidence* in predicting the true class of an instance and the *variability* of this confidence across epochs. To achieve this, multiple checkpoints are saved throughout a training epoch, and probability scores are derived for each data instance to calculate their training dynamics. By plotting the training dynamics of all instances on a data map, instances can be categorized into three groups: easy-to-learn, ambiguous, and hard-to-learn. For instance, consider a QA pair where a model consistently assigns a higher logit score to the correct answer than to other distractors across multiple checkpoints during an epoch. In this scenario, the model exhibits high confidence and low variability for that specific instance, suggesting that it is easy to learn. Conversely, instances with higher variability are ambiguous to the model, and those with low confidence are difficult to learn. Experimental results by Swayamdipta et al. (2020) demonstrates that training the model with ambiguous data contributes the most to out-of-distribution generalization.

Inspired by this finding, our research investigates the role of abstract commonsense knowledge within the training set and the effects of leveraging conceptualization. Since our QA model is trained with a marginal ranking loss, as described in Section 4.3, it does not output a probability

score but rather an MLM score for each option. Thus, the definition of model's *confidence* proposed by Swayamdipta et al. (2020) does not fit into our problem definition. To address this, we re-define the calculation of *confidence* to align with the model's degree of certainty in predicting an instance as the true class. Formally, denote $n$ as the number of saved checkpoints during an epoch for computing their training dynamics and the list of $m$ options in a $(Q_i, A_i)$ pair as $A_i = \{A_{i,1}, A_{i,2}, ..., A_{i,m}\}$ with $A_{i,j}$ being the ground-truth answer $(1 \leq j \leq m)$). We define the *confidence* of the model for such a QA pair in Equation 3, where $\sigma$ is the sigmoid function and $S_{i,d}^c$ is the score of option $A_{i,d}$ at checkpoint $c$.

$$\mathcal{C}(Q_i, A_i) = \frac{1}{n} \sum_{c=1}^{n} \sigma\left(\frac{\sum_{d=1}^{m}(S_{i,d}^c - S_{i,j}^c)}{m-1}\right) \quad (3)$$

Intuitively, this equation averages the gap between the ground-truth answer's score and the score of each distractor. A larger gap indicates a more confident model when choosing the answer. Variability aligns with the definition established by Swayamdipta et al. (2020). Specifically, it is calculated as the standard deviation of the score gap between the ground-truth answer and the distractors relative to the level of confidence exhibited throughout an entire epoch, as shown in Equation 4.

$$\mathcal{V}(Q_i, A_i) = \sqrt{\frac{\sum_{c=1}^{n}(\sigma(\frac{\sum_{d=1}^{m}(S_{i,d}^c - S_{i,j}^c)}{m-1}) - \mathcal{C}(Q_i, A_i))^2}{n}}$$
$$(4)$$

By revisiting the plots in Figure 4, we observe that the inclusion of abstract commonsense knowledge enhances the model's confidence and reduces variability when encountering knowledge in ATOMIC. The introduction of conceptualization appears to widen the differences between the model's predicted scores for the correct answer and those for the distractors. This suggests that the correct answer is more likely to be selected, leading to an improved learning outcome. However, the introduction of knowledge from ATOMIC-10X results in a reversed trend, indicating that it does not aid in better learning ATOMIC. Furthermore, we observe that abstract knowledge derived from conceptualizations is more ambiguous to the model in the conceptualization-augmented ATOMIC, which theoretically contributes more to out-of-domain generalization. Nonetheless, ATOMIC-10X still contains some easy-to-learn knowledge that does

| Models | aNLI | CSQA | PIQA | SIQA | WG |
|---|---|---|---|---|---|
| CAR (RoBERTa) | 72.7 | 66.3 | 73.2 | 64.0 | 62.0 |
| ◇ w/o CA | 72.3 | 64.8 | 73.2 | 64.8 | 61.3 |
| ◇ w/o CCQS | 71.5 | 67.3 | 72.1 | 61.8 | 62.7 |
| CAR (DeBERTa) | 79.6 | 69.3 | 78.6 | 64.0 | 78.2 |
| ◇ w/o CA | 78.9 | 67.2 | 78.6 | 63.8 | 78.1 |
| ◇ w/o CCQS | 78.2 | 68.1 | 78.1 | 63.5 | 78.3 |

Table 9: Ablation study on two components of CAR. CA stands for Conceptualization Augmentation, and CCQS stands for Concept-Constrained QA Synthesis. The following five columns denote the accuracy (%) on each benchmark.

not facilitate the model's generalization. Thus, abstract commonsense knowledge benefits zero-shot commonsense QA better than ATOMIC-10X by providing more ambiguous conceptual knowledge, which aids in making the model more generalizable.

We also plot the changes in training dynamics on different QA benchmarks, comparing models with and without the injection of abstract knowledge. The plots are shown in Figure 7. We observe that the inclusion of abstract commonsense knowledge significantly improves the models' confidence in downstream QA entries. However, the impact on the trend of variability is unclear. Nevertheless, this improvement in average confidence provides strong evidence for the model's enhancement in these downstream QA benchmarks.

## B.7 Ablation Study

Next, we study the ablation of different components in our CAR framework to determine the impact of utilizing conceptualization through various techniques. There are two critical components that distinguish CAR from traditional zero-shot QA systems (Ma et al., 2021):

• Conceptualization Augmentation: Augmenting the original commonsense knowledge in a CSKB with its conceptualizations to derive abstract commonsense knowledge. This knowledge is then synthesized into QA pairs, enabling the model to reason from a more generalized perspective. Without this component, abstract commonsense knowledge is not incorporated into the CSKB. Conceptualizations still remain as constraints for assisting QA pair synthesis, resulting in an approach that is similar to applying our proposed QA synthesis protocol directly to ATOMIC.

• Concept-Constrained QA Synthesis: Constraining a question's distractors by ensuring that none of their head events share a common keyword or conceptualization with the question's keywords and conceptualizations. If this component is dropped, the constraint will be downgraded, and only no sharing of common keywords between the question and distractors will be restricted. This approach introduces abstract commonsense knowledge into the CSKB and uses the original distractor generation strategy for synthesizing QA pairs.

We then train two batches of QA models, using RoBERTa-Large and DeBERTa-v3-Large as the backbone, by sequentially dropping the two components mentioned above one at a time. Their zero-shot performances on five commonsense QA benchmarks are reported in Table 9. From the results, it is observed that both components play important roles in CAR, with CCQS being more effective on average. This underscores the significance of eliminating false negative distractors, and conceptualization proves to be a useful tool for achieving this objective in improving the QA model's overall performance.

## B.8 The Effect of Conceptualization

Lastly, we study the improvement in the generalizability of our framework with the aid of conceptualizations by examining the accuracy gains on questions with varying levels of semantic overlap with knowledge in ATOMIC's training split. To do so, we sort the questions in every benchmark by their average BERTScore (Zhang et al., 2020) between each individual question entry against the whole training set in the original ATOMIC. We then split the questions into two sets based on their BERTScores, with the lower BERTScore indicating a lower semantic overlap and a greater need for the model to generalize to answer the question. These questions are denoted as "Difficult." Conversely, we refer to questions with high BERTScores as "Easy."

Then, we train two QA models following the pipeline proposed by Ma et al. (2021), with one trained on conceptualization-augmented ATOMIC and the other on ATOMIC only. We evaluate their performance on five commonsense QA benchmarks and compare the performance gains between two sets of questions in each benchmark, as shown in Figure 6. Results demonstrate that incorporating conceptualizations positively impacts accuracy,

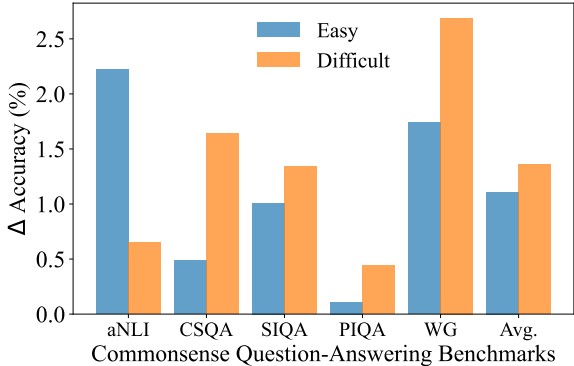

Figure 6: Comparison of accuracy improvement (%) with/without conceptualization-augmentation for two groups of QA entries across five benchmarks. Avg. stands for averaging across all benchmarks.

particularly for questions that deviate significantly from ATOMIC across multiple benchmarks. This indicates that augmenting ATOMIC with conceptualizations can improve the model's generalizability, particularly for questions that tend to be out-of-distribution, requiring more relevant knowledge to answer correctly.

## B.9 Generalization to Other CSKBs

While our work primarily experiments with the AbstractATOMIC dataset as the conceptualization source of ATOMIC, we also aim to extend our framework to other CSKBs for a more generalizable evaluation. To address this, we follow Ma et al. (2021) and explore the feasibility of transferring our framework to the CWWV dataset, which comprises multiple CSKBs including ConceptNet (Speer et al., 2017), WordNet (Miller, 1995), and WikiData (Vrandecic and Krötzsch, 2014). To accomplish this, we train a conceptualization generator based on GPT2 (Radford et al., 2019) and utilize ChatGPT (OpenAI, 2022) as two flexible generative conceptualizers. The generated conceptualizations are then transformed into abstract knowledge and integrated into the CWWV dataset. This augmented dataset is employed to train a zero-shot commonsense QA reasoner using our proposed CAR framework. We present the experimental results and compare them with baselines in Table 10. Our observations reveal a modest improvement in an average accuracy of 1% compared to all baselines and comparable performance to GPT3.5. These results demonstrate the effectiveness of incorporating conceptualizations from other CSKBs. In future research, we suggest exploring

| Model | CSKB | a-NLI | CSQA | PIQA | SIQA | WG | Avg. |
|---|---|---|---|---|---|---|---|
| RoBERTa-L (MR) (Ma et al., 2021) | CWWV | 70.0 | 67.9 | 72.0 | 54.8 | 59.4 | 64.8 |
| MTL (Kim et al., 2022) | CWWV | 69.6 | 67.3 | 72.5 | 52.0 | 57.2 | 63.7 |
| ZS-Fusion (Kim et al., 2022) | CWWV | 69.6 | 67.6 | 73.1 | 53.7 | 59.5 | 64.7 |
| CAR-RoBERTa-L (Ours) | CWWV$^C$ | 71.6 | 68.4 | 73.0 | 55.4 | 60.6 | 65.8 |
| GPT-3.5 (`text-davinci-003`) | N/A | 61.8 | 68.9 | 67.8 | 68.0 | 60.7 | 65.4 |
| ChatGPT (`gpt-3.5-turbo`) | N/A | 69.3 | **74.5** | 75.1 | 69.5 | 62.8 | 70.2 |

Table 10: Zero-shot evaluation results (%) on five commonsense question answering benchmarks by models trained on the CWWV dataset. CWWV$^C$ refers to the augmented CWWV dataset using generated conceptualizations from a trained GPT2 generator and ChatGPT.

automatic construction methods for conceptualization resources in other CSKBs and investigating their potential benefits for general commonsense reasoning.

## C  Case Study

In this section, we present case studies to demonstrate the effectiveness of CAR. First, we discuss cases that illustrate the power of conceptualization augmentation, as shown in Table 12. By transforming triples into abstract commonsense knowledge, we can introduce more general knowledge into the CSKB and improve its coverage. Moreover, the newly introduced triples were missing from the original CSKB. For instance, conceptualizing "PersonX plays the games together" as an "entertainment activity" introduces higher-level knowledge that cannot be simply represented by the original triple. Additionally, by synthesizing both types of triples into QA pairs, the QA model can learn both types of knowledge, which can help the model perform more generalizable reasoning on out-of-distribution commonsense QA benchmarks.

Next, in Table 13, we present QA pairs consisting of false negative options generated using keyword constraints during their synthesis from both the original ATOMIC and ATOMIC-10X. We also demonstrate how our concept constraint resolves this issue. Through these case studies, we observe that the original distractors may contain one or even both plausible options, which is suboptimal when training a QA model. Specifically, for distractors sampled from ATOMIC-10X, we observe that several distractors are vague and general (denoted as "?"), which can be plausible in many contexts. For example, in various triples, adjectives like "happy" and verb phrases such as "do it" are easy to be plausible and do not serve as significant distractions. This is not desirable when training a QA model. However, by using conceptualizations

as a constraint, the newly sampled distractors are all strong negatives, allowing the model to learn from such negative commonsense knowledge. This is because the distractors are sourced from triples that are more likely to be irrelevant to the original triple's context and, thus, more likely to be truly negative distractors.

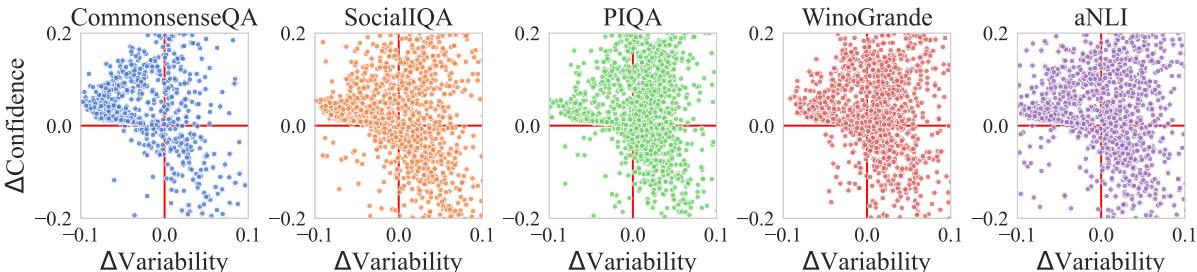

Figure 7: The change of training dynamics on various commonsense QA benchmarks by a DeBERTa-v3-Large model trained on abstract commonsense knowledge injected ATOMIC (ours) compared with the one trained only on ATOMIC (Ma et al., 2021).

| Model | CSKB | a-NLI | CSQA | PIQA | SIQA | WG | Avg. |
|---|---|---|---|---|---|---|---|
| Random | - | 50.0 | 20.0 | 50.0 | 33.3 | 50.0 | 40.7 |
| Majority | - | 50.8 | 20.9 | 50.5 | 33.6 | 50.4 | 41.2 |
| GPT2-L (Radford et al., 2019) | - | 56.5 | 41.4 | 68.9 | 44.6 | 53.2 | 52.9 |
| RoBERTa-L (Liu et al., 2019) | - | 65.5 | 45.0 | 67.6 | 47.3 | 57.5 | 56.6 |
| DeBERTa-v3-L (He et al., 2023) | - | 59.9 | 25.4 | 44.8 | 47.8 | 50.3 | 45.6 |
| Self-talk (Shwartz et al., 2020) | - | | 32.4 | 70.2 | 46.2 | 54.7 | - |
| COMET-DynGen (Bosselut et al., 2021) | ATOMIC | - | - | - | 50.1 | - | - |
| SMLM (Banerjee and Baral, 2020) | * | 65.3 | 38.8 | - | 48.5 | - | - |
| **Backbone: RoBERTa-Large** 340M | | | | | | | |
| RoBERTa-L (Vanilla) (Liu et al., 2019) | - | 65.5 | 45.0 | 67.6 | 47.3 | 57.5 | 56.6 |
| MICO (Su et al., 2022) | ATOMIC | - | 44.2 | - | 56.0 | - | - |
| RoBERTa-L (MR) (Ma et al., 2021) | ATM$_{10X}$ | 70.8 | 64.2 | 71.7 | 61.0 | 60.7 | 65.7 |
| RoBERTa-L (MR) (Ma et al., 2021) | ATOMIC | 70.8 | 64.2 | 72.1 | 63.1 | 59.2 | 65.9 |
| RoBERTa-L (MR) (Ma et al., 2021) | CWWV | 70.0 | 67.9 | 72.0 | 54.8 | 59.4 | 64.8 |
| RoBERTa-L (MR) (Ma et al., 2021) | CSKG | 70.5 | 67.4 | 72.4 | 63.2 | 60.9 | 66.8 |
| STL-PLM (Kim et al., 2022) | ATOMIC | 71.6 | 64.0 | 72.2 | 63.2 | 60.5 | 66.3 |
| MTL (Kim et al., 2022) | CWWV | 69.6 | 67.3 | 72.5 | 52.0 | 57.2 | 63.7 |
| MTL (Kim et al., 2022) | CSKG | 69.8 | 67.1 | 72.0 | 61.9 | 59.3 | 66.0 |
| STL-Adapter (Kim et al., 2022) | ATOMIC | 71.3 | 66.5 | 71.1 | 64.4 | 60.3 | 66.7 |
| STL-Adapter (Kim et al., 2022) | CSKG | 71.5 | 66.7 | 72.1 | 64.7 | 59.0 | 66.8 |
| ZS-Fusion (Kim et al., 2022) | CWWV | 69.6 | 67.6 | 73.1 | 53.7 | 59.5 | 64.7 |
| ZS-Fusion (Kim et al., 2022) | CSKG | 72.4 | 68.3 | 73.0 | 66.7 | 60.9 | 68.3 |
| MKIF (Guan et al., 2023) | CSKG | 72.5 | 71.0 | 73.1 | - | 61.0 | - |
| **CAR-RoBERTa-L (Ours)** | ATOMIC | 72.3 | 64.8 | 73.2 | 64.8 | 61.3 | 67.3 |
| **CAR-RoBERTa-L (Ours)** | ATM$^C$ | 72.7 | 66.3 | 73.2 | 64.0 | 62.0 | 67.6 |
| **Backbone: DeBERTa-v3-Large** 435M | | | | | | | |
| DeBERTa-v3-L (MR) (Ma et al., 2021) | ATM$_{10X}$ | 74.0 | 65.4 | 73.8 | 59.5 | 73.9 | 69.3 |
| DeBERTa-v3-L (MR) (Ma et al., 2021) | ATOMIC | 76.0 | 67.0 | 78.0 | 62.1 | 76.0 | 71.8 |
| **CAR-DeBERTa-v3-L (Ours)** | ATOMIC | 78.9 | 67.2 | **78.6** | 63.8 | 78.1 | 73.3 |
| **CAR-DeBERTa-v3-L (Ours)** | ATM$^C$ | **79.6** | 69.3 | **78.6** | 64.0 | **78.2** | **73.9** |
| **Large Language Models** | | | | | | | |
| GPT-3.5 (text-davinci-003) | - | 61.8 | 68.9 | 67.8 | 68.0 | 60.7 | 65.4 |
| ChatGPT (gpt-3.5-turbo) | - | 69.3 | **74.5** | 75.1 | **69.5** | 62.8 | 70.2 |
| **Supervised Learning & Human Performance** | | | | | | | |
| RoBERTa-L (Supervised) | - | 85.6 | 78.5 | 79.2 | 76.6 | 79.3 | 79.8 |
| DeBERTa-v3-L (Supervised) | - | 89.0 | 82.1 | 84.5 | 80.1 | 84.1 | 84.0 |
| Human Performance | - | 91.4 | 88.9 | 94.9 | 86.9 | 94.1 | 91.2 |

Table 11: Zero-shot evaluation results (%) on five commonsense question answering benchmarks with baselines trained on multiple CSKBs. The best results are **bold-faced**, and the second-best ones are underlined. ATM$^C$ stands for the ATOMIC with abstract commonsense knowledge injected and ATM$_{10X}$ stands for ATOMIC-10X (West et al., 2022). All baseline results are consistent with their original papers. CWWV refers to the combination of ConceptNet (Speer et al., 2017), VisualGenome (Krishna et al., 2017), WikiData (Vrandecic and Krötzsch, 2014), and WordNet (Miller, 1995). CSKG (Ilievski et al., 2021) consists of ATOMIC (Sap et al., 2019a) and CWWV.

| Original Triple | Original Synthetic QA | Conceptualized Triple | Conceptualized Synthetic QA |
|---|---|---|---|
| PersonX looks cute, oWant, asks PersonX on a date. | Wynne looks cute. As a result, others wanted to?
A: thank him.
B∗: ask Wynne on a date.
C: thank Wynne. | PersonX [*pretty*], oWant, asks PersonX on a date. | Wynne [*pretty*]. As a result, others wanted to?
A: thank him.
B∗: ask Wynne on a date.
C: thank Wynne. |
| PersonX sets a new record, xWant, accept the prize. | Ray sets a new record. As a result, Ray wanted to?
A: get to safety.
B∗: accept the prize.
C: send the email. | PersonX [*achievement*], xWant, accept the prize. | Ray [*achievement*]. As a result, Ray wanted to?
A: get to safety.
B∗: accept the prize.
C: send the email. |
| PersonX plays the games together, xNeed, find someone to play with. | Logan plays the game together. Before, Logan needed to?
A: know the framework.
B∗: find someone to play with.
C: wash the clothes. | [*entertaiment activity*], xNeed, find someone to play with. | [*entertaiment activity*]. Before, Logan needed to?
A: know the framework.
B∗: find someone to play with.
C: wash the clothes. |

Table 12: Case study of conceptualized triples and their synthesized QA pairs. Given an original triple from ATOMIC, we conceptualize the triple by replacing an instance with its [*plausible conceptualization*] to form a conceptualized triple. The conceptualized triples are then synthesized into QA pairs using the same ground-truth answer and distractors, sampled for the original triple, to train the QA model. ∗ indicates the ground-truth answer.

| Question | Distractor Sampling Strategy | Distractor | F.Neg. |
|---|---|---|---|
| Jamie makes Alex's breakfast. As a result, Jamie wanted to? (*eat with Alex.*) | Keyword (ATOMIC) | show off the food.
take them home. | ✓
× |
| | Keyword (ATOMIC-10X) | be a good person.
do it. | ✓
? |
| | Keyword + Concept | discuss the question.
get warm. | ×
× |
| Berkeley joins Aspen's party. As a result, others felt? (*looked up to, admired.*) | Keyword (ATOMIC) | enjoyed.
good. | ✓
✓ |
| | Keyword (ATOMIC-10X) | happy.
good. | ✓
✓ |
| | Keyword + Concept | upset for having to give up their keys.
sad. | ×
× |
| Cody builds Logan house. Before, Cody needed to? (*have construction material.*) | Keyword (ATOMIC) | have the knowledge.
help with preparation. | ✓
✓ |
| | Keyword (ATOMIC-10X) | start.
eat well. | ?
× |
| | Keyword + Concept | have been smoking weed.
be with others. | ×
× |
| Ash is at the mall with West's friends. Before, Ash needed to? (*drive his car.*) | Keyword (ATOMIC) | decide to go.
buy apple seeds. | ✓
× |
| | Keyword (ATOMIC-10X) | get prepared.
study the situation. | ?
× |
| | Keyword + Concept | tries to do it but can't.
fail the test. | ×
× |

Table 13: Case study of the false negative options in the original QA synthesis and how our proposed conceptualization-constraint resolves such an issue. The (*ground truth answer*) is appended below each question. Keyword represents only using keywords as the constraint for sampling distractor, while Concept refers to using both the keywords and conceptualizations. F.Neg. refers to whether the distractor is a false negative one or not.