# OpenReview forum: "CAR: Conceptualization-Augmented Reasoner for Zero-Shot Commonsense Question Answering"
_EMNLP/2023/Conference — EMNLP 2023 Findings_

### Official Review · Reviewer_hZ2z · 2023-07-26

**Soundness:** 4

**Excitement:**

4: Strong: This paper deepens the understanding of some phenomenon or lowers the barriers to an existing research direction.

**Paper Topic And Main Contributions:**

This paper proposed a new method for constructing synthetic data for zero-shot commonsense question answering. The previous method for populating synthetic QA pairs from commonsense knowledge graphs results in a large amount of false negatives. Consequently, training models on those data leads to sub-optimal performance. The authors propose to first get the conceptualized knowledge triple (getting the abstraction of the head concept/event while keeping the relation and tail the same). Then they followed the same synthetic QA generation pipeline, but the sampled distractors are filtered based on both the triple itself and the conceptualization of the triple (through keyword matching). In this way, the number of false negatives in the distractor can be significantly reduced. The authors conducted experiments by training RoBERTa and DeBERTa models on the resulting synthetic QA sets and evaluating on 5 commonsense QA benchmarks in a zero-shot fashion. The results show that the proposed method outperforms various baselines on all 5 tasks.

**Questions For The Authors:**

1. I have trouble understanding the difference between event vs instance vs concept, in table 4. Does event refers to the head of a triple, and instances refer to all heads in KG that share the same meaning, and concepts refer to all abstraction of the current head? It would be good to define these terms somewhere.
2. Have you tried to compute the human accuracy on the synthetic data? In Ma et al. 21, human accuracy is only around 80%, which is also a sign of false negatives in data. Would human accuracy be higher on your dataset?
3. Have you studied the impact of the size of the training set? If you downsample the augmented data to the original ATOMIC size (same as Ma et al. 21), what effect would it have on the model’s performance?

**Reasons To Accept:**

1. The proposed method is well-motivated and very intuitive. The issue of false negatives in synthetic data is very challenging to solve. By abstracting the higher-level concept out of the knowledge triple surface form, the false negatives can be more effectively detected and filtered out.
2. The results show that the model’s performance improved significantly by training on the newly generated synthetic data (for both models and on all 5 tasks).

**Reasons To Reject:**

1. The generalization of this method to other knowledge graphs is unclear. The most important step in the proposed pipeline, the conceptualization of the knowledge triples, requires the AbstractATOMIC dataset (He et al. 22). It would be interesting to see if there are any alternatives to get conceptualization of the knowledge triple without AbstractATOMIC, or if the proposed method works on other knowledge graphs such as ConceptNet.
2. Some experimental details are not well-explained, which might weaken the arguments of this paper. For example, (a) In section 6.1, human evaluations are used to understand the diversity of the augmented knowledge triples, I’m wondering how many annotators labeled each instance, and is the inter-annotator agreement computed? (b) In lines 482-483, the knowledge is randomly sampled from ATOMIC-10X, then in lines 493 - 494, these sampled knowledge are treated as augmentation when computing the diversity metric. In this case, the comparison in terms of diversity and coverage might be unfair because different random samples of the data could lead to very different coverage of knowledge?
3. For some minor problems, please see questions

**Reproducibility:**

5: Could easily reproduce the results.

**Reviewer Confidence:**

4: Quite sure. I tried to check the important points carefully. It's unlikely, though conceivable, that I missed something that should affect my ratings.

---

> ### Author Rebuttal · Authors · 2023-08-28
>
> We appreciate your detailed feedback. The following paragraphs respond to the reasons-to-rejects together with the questions.
>
> # Generalization to Other CSKBs
>
> Thank you for addressing the issue of generalizability. In fact, there exist systematic approaches, other than large-scale annotation, to acquire conceptualization data for other CSKBs.
>
> For example, in our paper, we leveraged **a GPT2 conceptualization generator that was trained on AbstractATOMIC**. This generator is capable of generating conceptualizations for specific instances by considering the instance itself and its source event (*refer to our definitions below*). This powerful tool can be effectively utilized to **generate conceptualized triples for CSKBs beyond ATOMIC**. Furthermore, we recently discovered that **ChatGPT exhibits robust conceptualization capabilities**. By using few-shot prompts synthesized from AbstractATOMIC, we can **prompt ChatGPT to generate conceptualized triples based on any given knowledge in other CSKBs**.
>
> To address your concern, we have chosen **CWWV** (including ConceptNet, WordNet, and WikiData) as an additional dataset, distinct from ATOMIC, which was utilized by Ma et al. (2021) in their work. We employ the **GPT2 generator and ChatGPT to conceptualize the triples in CWWV**. Due to insufficient time during the response period, we are unable to conduct large-scale annotations for verification and evaluation. Thus, we prepared a set of case studies for your review.
>
> | Event                                                              | Instance                                       | GPT2 Conceptualization | ChatGPT Conceptualization |
> |--------------------------------------------------------------------|------------------------------------------------|------------------------|---------------------------|
> | You are likely to find a basketball court in high school gymnasium | basketball court                               | sport venue            | sport arena               |
> | You are likely to find a basketball court in high school gymnasium | high school gymnasium                          | gym                    | gym                       |
> | You are likely to find an electric socket in electrician's truck   | electric socket                                | plugs                  | Power sockets             |
> | electric power distribution grid is a facility                     | electric power distribution grid is a facility | facility               | base facility             |
> | An activity pilots can do is fly a jet                             | pilot                                          | driver                 | guidance                  |
> | An activity pilots can do is fly a jet                             | fly a jet                                      | control                | navigation                |
> | a pillow case is for holding items                                 | pillow case                                    | cover                  | pillow protector          |
>
> These case studies demonstrate that **small generative models, when fine-tuned on conceptualization data, and large language models can both be effective in conceptualizing other CSKBs**. Notably, large language models can achieve this without relying on AbstractATOMIC. We then **incorporate these conceptualized triples, referred to as AbstractCWWV, into the traditional QA synthesis** pipeline introduced by Ma et al. (2021). **A QA model, based on RoBERTa-Large, is trained on the QA pairs synthesized from CWWV + AbstractCWWV directly using the random distractor sampling strategy**. The results from the evaluation on five QA benchmarks, in comparison with the baselines, are presented below.
>
> | Method            | CSKB                | aNLI | CSQA | PIQA | SIQA | WG   | Avg. |
> |-------------------|---------------------|------|------|------|------|------|------|
> | Ma et al. (2021)  | CWWV                | 70.0 | 67.9 | 72.0 | 54.8 | 59.4 | 64.8 |
> | Kim et al. (2022) | CWWV                | 69.6 | 67.6 | 73.1 | 53.7 | 59.5 | 64.7 |
> | **Our RoBERTa-Large** | CWWV + AbstractCWWV | 71.6 | 68.4 | 73.0 | 55.4 | 60.6 | **65.8** |
>
> Due to time constraints during the response period, we were unable to reproduce CAR with CWWV+AbstractCWWV, and there was a lack of a critic model or human evaluation for filtering the generations. As a result, **we strongly believe that the current model and results are not fully optimized**. However, we did observe a **modest improvement of approximately 1% in accuracy compared with both baselines**. Therefore, **we are confident that there are potential avenues for extending our framework to other CSKBs with ease**. For instance, leveraging a fine-tuned GPT2 or other large language models. This would eliminate the need for the AbstractATOMIC dataset, as alternative resources can be constructed for other CSKBs automatically. By obtaining these resources, **our framework can be directly applied to other CSKBs, enabling the training of more robust and knowledgeable QA models**.
>
> In addition, future research can explore conceptualizations distillations from large language models. The constructed resource can be utilized to benefit downstream tasks, such as COMET (Bosselut et al., 2019), and enable zero-shot commonsense QA.
>
> # Unclear Experiment Details
>
> We are sorry that some details aren’t made clear right now.
>
> (a) To conduct the expert evaluation, **we carefully selected three annotators who possess considerable research experience in the field of commonsense**. These annotators are either undergraduate or graduate students and have actively participated in the publication of research papers. Prior to the evaluation, we provided them with thorough instructions on the task of conceptualization and all related annotation tasks. Moreover, they underwent pilot rounds to ensure their annotation abilities, and they achieved a minimum accuracy of 80% on a meticulously curated gold set comprising 20 annotation questions, handpicked by the authors of this paper. Subsequently, they were provided with comprehensive explanations elucidating the underlying rationales for any incorrect responses they provided during the pilot rounds. This process ensured that they possess a clear understanding of their mistakes and the associated reasoning behind them.
>
> Overall, our annotators demonstrated a high level of agreement among themselves, **with an inner annotation agreement (IAA) of 83% in terms of pairwise agreement. The Fleiss Kappa score, which measures inter-rater reliability, was calculated to be 0.64**. This score is comparable to the Fleiss Kappa score (0.62) reported by Ma et al. (2021) and **reflects substantial agreement among the annotators**. Consequently, we are confident in the **reliability and validity of our evaluations**.
>
> (b) The ATOMIC-10X augmentation undergoes filtering based on the critical value and is then sampled to ensure fairness among different augmentation methods. This process guarantees that the augmented knowledge aligns with the conceptualization in terms of quantity. Although the randomness introduced by this sampling might raise concerns regarding diversity and coverage, **we argue that it does not have a substantial impact**. This is primarily due to the **relatively large size of our sampled subset, which constitutes approximately 20% of the complete ATOMIC-10X dataset**. Consequently, each random sample contains a diverse range of knowledge that maintains a comparable level of diversity and coverage to the original ATOMIC dataset.
>
> To further validate this assertion, we conducted three additional sampling iterations and calculated the automatic diversity measure (insufficient time to re-conduct expert evaluation three times). **The resulting diversities were 33.8%, 36.4%, and 34.4%, averaging at 34.9%, which closely aligns with the diversity reported in our paper (35.6%).** Hence, we are confident that the **random sampling approach does not yield significant differences in terms of diversity and coverage** when compared to ATOMIC.
>
> # Event vs. Instance vs. Concept
>
> We base our approach on the definitions proposed by He et al. (2022) and Wang et al. (2023) for formulating conceptualizations of commonsense knowledge.
>
> **Events:** Each event represents a commonly observed occurrence that encompasses valuable subsequential or inferential commonsense knowledge. In AbstractATOMIC, the events are the head events of all triples in ATOMIC without a wildcard (‘_’).
>
> **Instances:** Within each event, multiple *instances* have been identified with semantic parsing tools, representing specific components of the event that are worthy of conceptualization.
>
> **Concepts:** Concepts are the conceptualization of each instance. These concepts are thus extracted from Probase/WordNet and further validated by human annotators or critic filtering models.
>
> To summarize, the following definitions can be established: For an event $e$, which is the head of an ATOMIC triple, an instance refers to either an entity within the event or the complete event itself. Multiple instances can exist within a single event, denoted as ${i_1, i_2, i_3, …, i_n} \in e$. A concept corresponds to the conceptualization of an instance, and multiple conceptualizations can be associated with a single instance, as demonstrated by $(i_1, c_1,1), (i_1, c_1,2), (i_1, c_1,3),..., (i_2, c_2,1), ..., (i_n, c_n,1), …, (i_n, c_n,m)$.
>
> For instance, consider the event "PersonX is drunk when exiting the bar." In this case, two instances can be identified: "PersonX is drunk when exiting the bar" and "bar." The conceptualization for the instance "PersonX is drunk when exiting the bar" may include "drunk" or "enjoyed," while the instance "bar" can be conceptualized as an "entertainment place" or a "fun place."
>
> # Synthetic QA Accuracy/Plausibility
>
> We have also conducted expert annotations of the synthetic QA set to determine the synthetic QA pairs’ accuracies. We only select QA pairs that are synthesized from triples in the original ATOMIC for fair comparison. The results indicate that the expert annotator selects the correct answer for the question with an accuracy of approximately **86%** among 300 sampled QA pairs. This accuracy **surpasses that of the synthesis (78%) conducted by Ma et al. (2021)**.
>
> Despite this improvement, we did not include this information in our paper. The reason for this omission is that the accuracy is actually limited (more precisely, upper bounded) by the plausible rate of knowledge in ATOMIC, which is hard to quantify exactly. Given that certain knowledge within ATOMIC is challenging to comprehend, the corresponding QA pairs generated from this knowledge tend to be difficult to answer correctly. Ma et al. (2021) reported this issue as “Grammarly Incorrect,” which we also observed in our annotation. Consequently, we have opted to employ the **false-negative ratio as the main evaluation metric** to reflect this issue. This offers a more direct assessment and remains **unaffected by implausible or ungrammatical knowledge in ATOMIC**. Our expert evaluations demonstrate that **our proposed method effectively minimizes the likelihood of generating false-negative distractors**.
>
> # Impact of Training Data Size
>
> Thank you for raising the question regarding the size of training data. To answer your question, **we add experiments by downsampling the augmented training data (synthetic QA pairs from both ATOMIC and AbstractATOMIC) to the number of QA pairs synthesized only from ATOMIC**. DeBERTa-v3-Large is used as the backbone, and the results are reported in the table below:
>
> | Method                              | CSKB                  | aNLI | CSQA | PIQA | SIQA | WG   | Avg. |
> |-------------------------------------|-----------------------|------|------|------|------|------|------|
> | Baseline (Ma et al., 2021)          | ATOMIC                | 76.0 | 67.0 | 78.0 | 62.1 | 76.0 | 71.8 |
> | CAR                                 | ATOMIC                | 78.9 | 67.2 | 78.6 | 63.8 | 78.1 | 73.3 |
> | CAR **(Downsample to Half ATOMIC)** | ATOMIC+AbstractATOMIC | 77.3 | 64.0 | 75.6 | 61.0 | 76.4 | 70.9 |
> | CAR **(Downsample to ATOMIC)**      | ATOMIC+AbstractATOMIC | 79.2 | 67.7 | 77.0 | 61.9 | 77.5 | 72.7 |
> | CAR **(Without Downsampling)**      | ATOMIC+AbstractATOMIC | 78.5 | 69.3 | 78.1 | 63.9 | 78.1 | 73.6 |
>
> Upon downsampling the synthesized QA pairs obtained from a combination of ATOMIC and AbstractATOMIC to match the quantity of those derived solely from ATOMIC, we observed an average performance decrease of 0.9%. This decline resulted in the model's performance falling below that of CAR, which operates solely on ATOMIC without any conceptualization augmentation. Moreover, when downsampling to even half the quantity of ATOMIC QA pairs, the performance dropped significantly, falling below the baseline established by Ma et al. (2021).
>
> Given that conceptualization augmentation introduces a substantial amount of abstract commonsense knowledge, such downsampling leads to the loss of a considerable portion of both abstract knowledge (from AbstractATOMIC) and concrete knowledge (from ATOMIC). Therefore, we can conclude that **both types of knowledge are crucial for developing a robust zero-shot commonsense QA model**. While augmenting ATOMIC with conceptualization proves advantageous to the model, **it still necessitates a substantial amount of data for the model to exhibit generalizability and optimal performance**.
>
> # References
>
> He, M., Fang, T., Wang, W., & Song, Y. (2022). Acquiring and Modelling Abstract Commonsense Knowledge via Conceptualization. ArXiv, abs/2206.01532.
>
> Ma, K., Ilievski, F., Francis, J.M., Bisk, Y., Nyberg, E., & Oltramari, A. (2020). Knowledge-driven Data Construction for Zero-shot Evaluation in Commonsense Question Answering. AAAI Conference on Artificial Intelligence.
>
> Wang, W., Fang, T., Xu, B., Bo, C.Y., Song, Y., & Chen, L. (2023). CAT: A Contextualized Conceptualization and Instantiation Framework for Commonsense Reasoning. Annual Meeting of the Association for Computational Linguistics.
>
> Bosselut, A., Rashkin, H., Sap, M., Malaviya, C., Celikyilmaz, A., & Choi, Y. (2019, July). COMET: Commonsense Transformers for Automatic Knowledge Graph Construction. In Proceedings of the 57th Annual Meeting of the Association for Computational Linguistics (pp. 4762-4779).
>
> ***We assure you that we will include all the mentioned content in our final camera-ready version to improve the clarity and readability of our paper. We sincerely appreciate your valuable advice and hope that our response will assist you in raising your score. Thank you once again!***

---

### Official Review · Reviewer_NWmS · 2023-08-03

**Soundness:** 4

**Excitement:**

3: Ambivalent: It has merits (e.g., it reports state-of-the-art results, the idea is nice), but there are key weaknesses (e.g., it describes incremental work), and it can significantly benefit from another round of revision. However, I won't object to accepting it if my co-reviewers champion it.

**Paper Topic And Main Contributions:**

The paper aims to solve zero-shot commonsense QA tasks. It constructs synthetic QA pairs based on existing knowledge triplets for pre-training a masked language model. To improve coverage and reduce false negatives, the work adopts knowledge abstraction. This helps: 1) augment the commonsense KG to an abstract one and 2) filter false negatives (tail entities). During inference, the trained LM is directly applied to conduct QA without further tuning. Experiments show effectiveness of the proposed method on a wide range of commonsense QA tasks.

**Reasons To Accept:**

1.	The proposed idea, i.e., using knowledge abstraction to improve coverage and reduce false negatives, is sound and novel.
2.	The proposed method brings significant performance gain without direct supervision.
3.	The paper is well-organized and easy to follow.

**Reasons To Reject:**

1.	I understand that a multi-choice QA setting would facilitate evaluation. But I do not think it still makes sense today to assume there exist options to choose from when conducting commonsense reasoning.
2.	It is unclear why training LMs on synthetic QA pairs based on knowledge triplets can lead to downstream performance gain. Tasks like PIQA and SIQA have long context which is not simply about a head entity and relation. Also, certain questions require multi-hop reasoning. It would be helpful if the work can provide some explanations on the impact of pre-training.

**Reproducibility:**

4: Could mostly reproduce the results, but there may be some variation because of sample variance or minor variations in their interpretation of the protocol or method.

**Reviewer Confidence:**

3: Pretty sure, but there's a chance I missed something. Although I have a good feel for this area in general, I did not carefully check the paper's details, e.g., the math, experimental design, or novelty.

---

> ### Author Rebuttal · Authors · 2023-08-26
>
> We appreciate your detailed feedback. The following respond to the reasons-to-reject.
>
> # The Necessity and Importance of Multi-Choice Question Answering (MCQA)
>
> First, the inclusion of negative options is essential as it acknowledges **the subjective and comparative nature of commonsense knowledge**. Rather than assuming the question-answer pair as an absolute truth, **Multiple-Choice Question Answering (MCQA) considers the choices as plausible, with minimal cost of incorporating bridging inferences in the causal field** (COPA; Roemmele et al., 2011).
>
> For instance, in certain scenarios, even negative options can be deemed plausible when presented with counterfactual context. For example, in zero gravity, an apple may float in the air instead of falling down. However, when answering a commonsense reasoning question, it is prudent to compare it against the most common case without imposing additional restrictions. Hence, when considering the question of what happens when an apple ripens, selecting "apples fall when ripe" as a plausible choice is more appropriate than choosing "apples float in the air when ripe." To this extent, in order to evaluate a commonsense reasoning model's ability to compare and distinguish the most plausible knowledge in typical scenarios that are commonly recognized, it is **beneficial to incorporate negative options as comparative benchmarks to form an efficient evaluation** approach.
>
> Second, practically speaking, **MCQA serves as the predominant representation for commonsense reasoning datasets, encompassing a wide range of benchmarks that explore various aspects of commonsense**. These include SWAG, HellaSWAG, CommonsenseQA (v1.0, v2.0), SocialIQA, PhysicalIQA, ARC, OpenbookQA, CosmosQA, WinoGrande, COPA, CODAH, MC-TACO, a-NLI, among others (citations omitted). It is highly debatable to argue that the previous constructions and subsequent works in these commonsense QA datasets have become obsolete, as solving these tasks remains **a crucial objective even for large language models**.
>
> # Impact of Pre-training and Knowledge Injection
>
> Thank you for your attention to the impact of pre-training. We would like to provide further clarification on this approach, which involves injecting knowledge into a model that is similar to transfer learning. By using synthetic question-answering pairs from commonsense knowledge bases, we enable **the model to become familiar with specific types of question-answering data and grasp a wide range of commonsense knowledge encapsulated within the CSKBs and conceptualized CSKBs**. This method effectively **transfers commonsense knowledge into the model**, as demonstrated in studies by Ma et al. (2021), Bosselut et al. (2021), and Shwartz et al. (2020) (self-talk). Notably, in our experiments, models perform exceptionally well on SocialIQA, a commonsense QA dataset derived from ATOMIC, the CSKB used for pre-training. This further demonstrates the model's ability to transfer knowledge from CSKBs to answer downstream QA tasks.
>
> It is also important to highlight that the synthetic QA pairs used in pre-training are **not limited to head, entity, and relation only**. For instance, QA pairs derived from ATOMIC predominantly feature an event in the question and an adjective/event in the answer. The objective of this pre-training approach involves comparing the loss between the correct choice and distractors, thereby **training the model to better discriminate plausible commonsense assertions and knowledge**. Such pre-training enables the model to effectively differentiate among diversified scenarios and identify the most plausible commonsense option. Additionally, datasets involving longer contextual questions or multi-hop reasoning, such as PIQA and SIQA, also benefit from this type of pre-training. As shown by our experiments, **the model’s improved sensitivity also generalizes well to these complex reasoning scenarios**.
>
> # References
>
> Roemmele, M., Bejan, C. A., & Gordon, A. S. (2011, March). Choice of plausible alternatives: An evaluation of commonsense causal reasoning. In 2011 AAAI Spring Symposium Series.
>
> Ma, K., Ilievski, F., Francis, J.M., Bisk, Y., Nyberg, E., & Oltramari, A. (2020). Knowledge-driven Data Construction for Zero-shot Evaluation in Commonsense Question Answering. AAAI Conference on Artificial Intelligence.
>
> Bosselut, A., Le Bras, R., & Choi, Y. (2021, May). Dynamic neuro-symbolic knowledge graph construction for zero-shot commonsense question answering. In Proceedings of the AAAI conference on Artificial Intelligence (Vol. 35, No. 6, pp. 4923-4931).
>
> Shwartz, V., West, P., Le Bras, R., Bhagavatula, C., & Choi, Y. (2020, November). Unsupervised Commonsense Question Answering with Self-Talk. In Proceedings of the 2020 Conference on Empirical Methods in Natural Language Processing (EMNLP) (pp. 4615-4629).
>
> ***We sincerely appreciate your valuable advice and hope that our response will assist you in raising your score. Thank you once again!***

---

### Official Review · Reviewer_qJZg · 2023-08-04

**Soundness:** 4

**Excitement:**

3: Ambivalent: It has merits (e.g., it reports state-of-the-art results, the idea is nice), but there are key weaknesses (e.g., it describes incremental work), and it can significantly benefit from another round of revision. However, I won't object to accepting it if my co-reviewers champion it.

**Paper Topic And Main Contributions:**

This paper proposes a new solution for zero-shot commonsense question answering task via commonsense knowledge bases (CSKB)-based data augmentation. It proposes to leverage conceptualization-augmented CSKB to enlarge the range of synthesized QA instances and reduce false-negative distractors. Experiments show that the overall framework performs better than previous zero-shot baselines, and the conceptualization-augmented CSKB is a better than a CKSB distilled from GPT-3 as the source for learning general reasoning.

**Questions For The Authors:**

As Large language models (e.g. GPT-3.5-turbo) can directly be used for zero-shot commonsense reasoning and the performance can be further promoted with tricks like chain-of-thoughts, what is the main advantage of building new conceptualization resources for zero-shot commonsense reasoning?

**Reasons To Accept:**

1. This work provides extensive results of experiments and analyses, which shows the advantage of CSKB conceptualization for zero-shot commonsense question answering.

2. The result also brings some insight about the weakness of automatically generated commonsense knowledge bases.

**Reasons To Reject:**

1. The novelty of the method is weak. This work hugely relies on previous work on the conceptualization-augmented CSKB resource. The conceptualization (Section 4.1) is actually done in previous work (He et al. (2022) and Wang et al. (2023)). And the synthesizing QA method (Section 4.2) is almost the same as Ma et al. (2021) except for extend the negative-sampling constrains to the augmented resource. Therefore, the real contribution of this paper is relatively weak.

Mutian He, et al. 2022. Acquiring and modelling abstract commonsense knowledge via conceptualization.

Weiqi Wang, et al. 2023. CAT: A contextualized conceptualization and instantiation framework for commonsense reasoning.

Kaixin Ma, et al. 2021. Knowledge-driven data construction for zero-shot evaluation in commonsense question answering.

**Reproducibility:**

4: Could mostly reproduce the results, but there may be some variation because of sample variance or minor variations in their interpretation of the protocol or method.

**Reviewer Confidence:**

5: Positive that my evaluation is correct. I read the paper very carefully and I am very familiar with related work.

---

> ### Author Rebuttal · Authors · 2023-08-26
>
> We appreciate your detailed feedback. The following paragraphs respond to the reason-to-reject together with the question.
>
> # Weak Novelty
>
> Thank you for raising the issue of novelty. While existing frameworks, such as the conceptualization framework by He et al. (2022) and the data synthesis framework by Ma et al. (2021), have laid the foundation, our paper introduces novel contributions in two key areas. First, we present **a conceptualization-guided negative sampling strategy** that effectively reduces the occurrence of false-negative examples. Second, we propose **a new general data augmentation technique** based on conceptualization, accompanied by **comprehensive analyses of the data attribution of conceptualized data** for reliable usage.
>
> In terms of the negative sampling strategy (Section 4.2), we go beyond conceptualization augmentation alone. We suggest utilizing conceptualization as an extended constraint for negative sampling, ensuring that the sampled negative options have no common keywords or conceptualizations with those mentioned in the question. This novel approach to negative sampling holds promise for its **applicability to other tasks that require negative sampling**, such as CSKB Completion (Malaviya et al., 2019) and Population (Fang et al., 2021).
>
> Second, previous works on conceptualization have primarily focused on intrinsic evaluations using proprietary datasets without exploring their utility in downstream applications. Adapting conceptualization for practical downstream scenarios represents a significant and non-trivial advancement. By introducing conceptualization as **a general data augmentation tool**, we pave the way for new avenues of research. Notably, our augmented data demonstrate even greater generalizability compared to data distilled from GPT3 (ATOMIC-10X).
>
> In addition, our work is strengthened, made more interpretable, and innovative by conducting a thorough analysis of the properties of conceptualization augmented data (Section 6). Rather than simply applying conceptualization, we extensively analyze the data attribution of conceptualized augmented data, considering factors such as diversity, false-negative rate, and training dynamics.
> We demonstrate that conceptualization leads to the generation of more *ambiguous* (defined in Swayamdipta et al., 2020, a type of data useful for training) training examples, which significantly aids in the generalization of the model for performing commonsense reasoning. This observation is consistent with the findings in the training dynamics paper by Swayamdipta et al. (2020). For instance, we offer an explanation for why **conceptualization produces superior augmented data compared to that derived from GPT-3** (ATOMIC-10x). The reason lies in the ability of conceptualization to provide more ambiguous training signals, thereby enhancing the model's robustness.
>
> # Advantages of Conceptualization over Advanced LLM Prompting
>
> Thank you for considering the comparison between conceptualization and advanced LLM prompting techniques, such as CoT reasoning. We believe that there are three distinct advantages that highlight the superiority of conceptualization and our system.
>
> Firstly, our framework offers models that are **significantly smaller in scale** while maintaining comparable performance to LLMs like ChatGPT. For instance, our trained model, which is based on DeBERTa-v3-large, consists of just 400 million parameters, representing a mere 0.2% of a standard 175 billion parameter GPT-3 model. Despite its smaller size, our model surpasses ChatGPT in a purely zero-shot setting, showcasing its efficiency in both training and deployment. In contrast, advanced prompting techniques require extensive computational resources for deployment and operation. This feature makes **our model, trained with conceptualization, more versatile and accessible to a broader range of researchers in NLP community** who may lack the necessary access or resources to extensively prompt LLMs.
>
> Furthermore, conceptualization provides a broader range of commonsense knowledge, which is currently lacking in current CSKBs. Wang et al. (2023) have previously demonstrated the advantages of integrating this type of knowledge into commonsense inference generation (COMET; Bosselut et al., 2019). It is evident from this research that conceptualization serves as a valuable resource for training more resilient and dependable generative models. Consequently, the development of conceptualization resources holds great potential for **enhancing the performance of large language models in commonsense reasoning**, such as fine-tuning ChatGPT using conceptualization data.
>
> Lastly, we envision that **conceptualization data introduces the potential for a more advanced prompting of Language Model Models (LLMs)**. As we have emphasized previously, the process of conceptualization and instantiation of knowledge plays a crucial role in reasoning. By incorporating the "chain of concept" reasoning within the "chain of thought" paradigm, we can employ conceptualization as an additional guiding mechanism to prompt LLMs. Exploring this direction could bridge a significant gap in current understanding and yield promising future improvements in LLM performance across various downstream tasks.
>
> # References
>
> Malaviya, C., Bhagavatula, C., Bosselut, A., & Choi, Y. (2019). Commonsense Knowledge Base Completion with Structural and Semantic Context. AAAI Conference on Artificial Intelligence.
>
> Fang, T., Wang, W., Choi, S., Hao, S., Zhang, H., Song, Y., & He, B. (2021). Benchmarking Commonsense Knowledge Base Population with an Effective Evaluation Dataset. Conference on Empirical Methods in Natural Language Processing.
>
> Swayamdipta, S., Schwartz, R., Lourie, N., Wang, Y., Hajishirzi, H., Smith, N.A., & Choi, Y. (2020). Dataset Cartography: Mapping and Diagnosing Datasets with Training Dynamics. Conference on Empirical Methods in Natural Language Processing.
>
> He, M., Fang, T., Wang, W., & Song, Y. (2022). Acquiring and Modelling Abstract Commonsense Knowledge via Conceptualization. ArXiv, abs/2206.01532.
>
> Ma, K., Ilievski, F., Francis, J.M., Bisk, Y., Nyberg, E., & Oltramari, A. (2020). Knowledge-driven Data Construction for Zero-shot Evaluation in Commonsense Question Answering. AAAI Conference on Artificial Intelligence.
>
> Bosselut, A., Rashkin, H., Sap, M., Malaviya, C., Celikyilmaz, A., & Choi, Y. (2019, July). COMET: Commonsense Transformers for Automatic Knowledge Graph Construction. In Proceedings of the 57th Annual Meeting of the Association for Computational Linguistics (pp. 4762-4779).
>
> Wang, W., Fang, T., Xu, B., Bo, C.Y., Song, Y., & Chen, L. (2023). CAT: A Contextualized Conceptualization and Instantiation Framework for Commonsense Reasoning. Annual Meeting of the Association for Computational Linguistics.
>
> ***We sincerely appreciate your valuable advice and hope that our response will assist you in raising your score. Thank you once again!***

---

### Meta-Review · Area_Chair_6j9H · 2023-09-19

**Recommendation:** 5

**Metareview:**

This paper proposes a new solution to gather negative synthetic data to train an LM for zero-shot commonsense QA. For this, they use conceptualization/ abstraction over Commonsense KB triples to enhance knowledge coverage and in guiding an effective negative sampling strategy to mitigate false-negatives in the generated synthetic data. Their method gets empirical gains over data distilled directly from LLMs and they provide insightful analysis for on-the-fly generated CKB. This is a useful framework with good results, backed by good experimental analysis.

It would be interesting to see the generality of their conceptualization based pretrained LM by studying if it applies to commonsense generation tasks such as script generation (it should be within the scope as they use ATOMIC). For generalization of the approach w.r.t. other KBs, please include the new results in the camera ready. It would also be interesting to see how conceptualization can be used to guide LLM prompts.

In summary, this is a useful data augmentation technique that can yield good results on other tasks in the future (remains to be seen).

---

### Decision · Program_Chairs · 2023-10-07

**Decision:**

Accept-Findings

**Comment:**

This paper proposes a new solution to gather negative synthetic data to train an LM for zero-shot commonsense QA. For this, they use conceptualization/ abstraction over Commonsense KB triples to enhance knowledge coverage and in guiding an effective negative sampling strategy to mitigate false-negatives in the generated synthetic data. Their method gets empirical gains over data distilled directly from LLMs and they provide insightful analysis for on-the-fly generated CKB. This is a useful framework with good results, backed by good experimental analysis.

It would be interesting to see the generality of their conceptualization based pretrained LM by studying if it applies to commonsense generation tasks such as script generation (it should be within the scope as they use ATOMIC). For generalization of the approach w.r.t. other KBs, please include the new results in the camera ready. It would also be interesting to see how conceptualization can be used to guide LLM prompts.

In summary, this is a useful data augmentation technique that can yield good results on other tasks in the future (remains to be seen).